# Mechanistic Insights into the Pharmacological Significance of Silymarin

**DOI:** 10.3390/molecules27165327

**Published:** 2022-08-21

**Authors:** Karan Wadhwa, Rakesh Pahwa, Manish Kumar, Shobhit Kumar, Prabodh Chander Sharma, Govind Singh, Ravinder Verma, Vineet Mittal, Inderbir Singh, Deepak Kaushik, Philippe Jeandet

**Affiliations:** 1Department of Pharmaceutical Sciences, Maharshi Dayanand University, Rohtak 124001, Haryana, India; 2Institute of Pharmaceutical Sciences, Kurukshetra University, Kurukshetra 136119, Haryana, India; 3M.M. College of Pharmacy, Maharishi Markandeshwar (Deemed to be University), Ambala 133207, Haryana, India; 4Department of Pharmaceutical Technology, Meerut Institute of Engineering and Technology (MIET), Meerut 250005, Uttar Pradesh, India; 5Department of Pharmaceutical Chemistry, Delhi Pharmaceutical Sciences and Research University, New Delhi 110017, Delhi, India; 6Department of Pharmacy, G.D. Goenka University, Sohna Road, Gurugram 122103, Haryana, India; 7Chitkara College of Pharmacy, Chitkara University, Rajpura 140401, Punjab, India; 8Research Unit-Induced Resistance and Plant Bioprotection, University of Reims, EA 4707-USC INRAe 1488, SFR Condorcet FR CNRS 3417, 51687 Reims, France

**Keywords:** anti-inflammatory, antioxidant, pharmacological interventions, pro-apoptotic, silybin, silymarin

## Abstract

Medicinal plants are considered the reservoir of diverse therapeutic agents and have been traditionally employed worldwide to heal various ailments for several decades. Silymarin is a plant-derived mixture of polyphenolic flavonoids originating from the fruits and akenes of *Silybum marianum* and contains three flavonolignans, silibinins (silybins), silychristin and silydianin, along with taxifolin. Silybins are the major constituents in silymarin with almost 70–80% abundance and are accountable for most of the observed therapeutic activity. Silymarin has also been acknowledged from the ancient period and is utilized in European and Asian systems of traditional medicine for treating various liver disorders. The contemporary literature reveals that silymarin is employed significantly as a neuroprotective, hepatoprotective, cardioprotective, antioxidant, anti-cancer, anti-diabetic, anti-viral, anti-hypertensive, immunomodulator, anti-inflammatory, photoprotective and detoxification agent by targeting various cellular and molecular pathways, including MAPK, mTOR, β-catenin and Akt, different receptors and growth factors, as well as inhibiting numerous enzymes and the gene expression of several apoptotic proteins and inflammatory cytokines. Therefore, the current review aims to recapitulate and update the existing knowledge regarding the pharmacological potential of silymarin as evidenced by vast cellular, animal, and clinical studies, with a particular emphasis on its mechanisms of action.

## 1. Introduction

Herbal medicaments have been commonly utilized as therapeutic moieties across the globe for therapy and management of a wide array of ailments. Despite the vast advancements in the current medicinal system, medicinal plants still play an imperative role in humans’ well-being [1]. From ancient times, numerous indigenous plants have been employed worldwide to treat various illnesses. Among the diversity of medicinal plants, *Silybum marianum*, one of the most primordial and systematically researched plants, has been widely employed from ancient times as a natural medication for various liver diseases and some digestive issues of the upper gastrointestinal tract [2]. Indeed, *S. marianum*, belonging to the Asteraceae/Compositae family, is commonly named ‘milk thistle’ because of the presence of milky white veins on its leaves, which on breakage liberate a milky sap [3]. It is an inhabitant of the Mediterranean but has also grown for centuries all through North Africa, Europe and the Middle East region. In India, this plant is commonly found in Jammu and Kashmir at 1800–2400 m [4]. Silymarin, the standardized seed extract of milk thistle, has been extensively utilized as a broad-spectrum medicinal herb for a very long time [5,6]. From an ethnopharmacological point of view, silymarin has been employed for more than two centuries as a herbal therapy for protecting the liver from varied toxic matters, treating hepatic damage and for treatment of hepatitis as well cirrhosis [7,8]. Silymarin has also been used as an antidote for insect stings, snake bites, mushroom poisoning and alcohol [9,10,11].

Chemically, silymarin is a polyphenolic flavonoid extract consisting of about 70–80% silymarin flavonolignans along with 20–35% fatty acids and several other polyphenolic components [12]. Amongst all flavonolignans, silibinin (silybin), (*2R,3R*)-3,5,7-trihydroxy-2-[(*2R,3R*)-3-(4-hydroxy-3-methoxyphenyl)-2-(hydroxymethyl)-2,3-dihydrobenzo [b][1,4]dioxin-6-yl]chroman-4-one (Figure 1), is the foremost active compound present in silymarin with almost 60–70% abundance and exists in the form of two diastereomers, silybin A and silybin B [13]. The pathways for silybin biosynthesis are somehow not clearly identified, but specific biomimetic syntheses assert that the coupling of taxifolin and coniferyl alcohol via peroxidase activity leads to silybin formation [14,15]. The other flavonolignans available in silymarin consist of isosilybin (5%), silychristin (20%) and silydianin (10%). As with silybin, isosilybin also naturally occurs in two diastereomeric forms, i.e., isosilybin A and isosilybin B (Figure 1). Some minor flavonolignans that are also present in silymarin include silimonin and isosilychristin [16]. Apart from flavonolignans, taxifolin is an essential flavonoid found in silymarin [17,18,19].

Evidence from preclinical and clinical research has revealed that silymarin and its flavonolignans significantly impart antioxidant, anti-inflammatory and pro-apoptotic properties, inducing numerous biological and pharmacological activities, for instance, hepatoprotection, neuroprotection, anti-diabetic properties, anti-cancer properties, cardioprotection, photoprotection, immunomodulation and many more. In the United States, there is no quality control and regulation of herbal compounds such as silymarin as they are not considered drugs and are not under the supervision of the US Food and Drug Administration [20]. Because of its excellent therapeutic efficacies, it is one of the most widely used dietary supplements and around 75 brands of silymarin are available on the market in different dosage forms (tablets, capsules, syrups, etc.) with enhanced bioavailability under trade names such as Livergol^®^, Silipide^®^, Carsil^®^ tablets, Legalon^®^ capsules and Alrin-B^®^ syrup. Nano silymarin OIC, approved by the Vietnam Drug Administration, is the only patented nanoformulation that is commercially available as a dietary supplement, in the form of capsules, for improving liver function [21,22,23].

Silymarin offers several benefits in contrast to other therapeutic agents because of its non-toxicity and excellent hydrophobic properties. It has low aqueous solubility which results in poor bioavailability. This issue can be resolved by employing a nanosystems-based approach. Nanoparticles are naturally or chemically synthesized particles that have a particle size range of 1–200 nm. Because of their small size range, they offer various advantages, such as an enhanced interaction area, enhanced aqueous solubility and intracellular permeability. In addition, they can reduce the multi-drug resistance of many anti-cancer agents, including silybin. These are distributed across the body depending upon various factors, such as their small size which aids in longer systematic circulation periods and their potential to take advantage of anti-cancer properties. They also show greater stability during storage. Nanoparticles and their use in drug delivery are a much more efficient approach for cancer treatment than traditional chemotherapy [24,25,26].

Elfaky et al. (2022) investigated the hepatoprotective potential of silymarin nanoparticles (NPs) with different particle size in Sprague Dawley adult male rats. They reported that large silver NPs are more effective in hepatoprotective action compared to small NPs [27]. Abdullah et al. (2022) designed a novel nanoformulation of silymarin-loaded chitosan NPs for improving anti-fibrotic potential against liver fibrosis. NPs were developed using the ionotropic gelation method. The authors reported that the developed formulation resulted in significant anti-fibrotic action against CCl_4_-induced hepatic injury [28]. Patel et al. (2022) also developed silybin-loaded NPs for inhalation with caprolactone/pluronic F68 for the treatment of lung cancer. Pharmacokinetic investigations have revealed that the developed NPs enhanced silybin bioavailability, with a more than 4-times increase in AUC in contrast to *i/v* administration [29]. In another study, Iqbal et al. (2022) engineered *Silybum marianum*-mediated biosynthesized copper oxide NPs and investigated their various biological activities, such as anti-microbial properties, catalytic properties, anti-diabetic properties, antioxidant properties and ROS/RNS inhibition. They concluded that the developed NPs have significant in vitro biological and biomedical activities. They can be employed as broad spectrum agents for various biomedical applications [30]. Finally, Staroverov et al. (2021) fabricated silymarin–selenium NP conjugates with 30–50 ± 0.5 nm particle size. The developed conjugates enhanced cellular dehydrogenase activity and facilitated its penetration into intracellular spaces [31].

It has also been reported that co-administration of silymarin with some therapeutic agents may enhance its biological activity that is reduced by the liver, such as amitriptyline, diazepam, celecoxib, fluvastatin, diclofenac, zileuton, ibuprofen, glipizide, losartan, irbesartan, piroxicam, tolbutamide, torsemide, tamoxifen and phenytoin [32]. Han et al. (2009) reported that the coadministration of different sources of silymarin with losartan significantly improved the systemic concentration of Losartan [33]. Similarly, Molto et al. (2012) examined the effect of co-administration of silymarin with darunavir and ritonavir combinations in HIV patients. They reported a decline in AUC and C_max_ when co-administration of silymarin was used in combination with drugs compared to combinations of the drugs alone [34].

Considering the potential properties of silymarin, the present review has been designed to provide an insight into of its numerous pharmacological activities with detailed information about its mechanisms of action.

## 2. Pharmacological Aspects of Silymarin

Silymarin possesses a tremendous array of biological and pharmacological potential by interacting directly or indirectly with several molecular targets, including transcription factors, inflammatory mediators, protein kinases, receptors and enzymes, as illustrated in Figure 2.

### 2.1. Hepatoprotective Activity

The liver is the vital organ for metabolism of xenobiotics, lipids and numerous environmental pollutants and helps eliminate many substances from the body [35]. Silymarin has a prolonged history of traditional use in Ayurvedic medicine as a hepatoprotective agent and is nowadays broadly employed in the treatment and management of numerous hepatic disorders such as alcoholic liver disease, hepatic cancers, non-alcoholic fatty liver disease (NAFLD), non-alcoholic steatohepatitis (NASH) and drug toxicity, as mentioned in Table 1. Numerous studies have been conducted on the potent hepatoprotective actions of silymarin and its flavonolignans in the last decade. The various proposed mechanisms through which silymarin exerts its hepatoprotective activity are represented in Figure 3. Initially, gamma-glutamyl transferase (γGT), glutamic-oxalacetic transaminase (SGOT), glutamic pyruvic transaminase (SGPT) and alkaline phosphate (ALP) are essential and characteristic enzymes of the liver, and their elevated levels indicate hepatotoxicity. Silymarin was found to decrease the levels of these hepatic enzymes and prevent cellular escape and loss of functional integrity of hepatocyte membranes. Additionally, silymarin and its flavonolignans have a significant role in the reduction of cholesterol (CH), triglyceride (TG) and low-density lipoprotein (LDL) levels along with elevation of the content of high-density lipoproteins (HDL) [36,37,38,39,40].

Oxidative stress has a critical position during the progression of NAFLD and hepatic steatosis [59], and the use of exogenous natural antioxidants such as silymarin can trigger various antioxidant enzymes and stimulate non-enzymatic nuclear factor erythroid 2-related factor 2 (Nrf2) pathways, which consequently diminishes oxidative stress [14,51,60]. Recently, Mengesha et al. (2021) evaluated the hepatoprotective effect of silymarin using a fructose-induced NAFLD rat model and concluded that silymarin significantly improves lipid profile and liver function along with amelioration of oxidative stress status [39]. Zhu et al. (2018) reported that *S. marianum* oil impedes oxidative stress in high-fat diet (HFD) rats by elevating the levels of endogenous antioxidant enzymes. Moreover, it was also observed that oral administration of this oil improves hepatic fatty acid synthesis and fatty acid oxidation by reducing the mRNA levels of the fatty acid synthase (FAS), the liver X receptor α and the sterol regulatory element-binding protein 1c (SREBP-1c) [50].

Along with oxidative stress, inflammation is considered to be another imperative mediator of NAFLD and NASH. Preclinical and clinical substantiations revealed that silymarin demonstrates anti-inflammatory actions via repressing the release of cytokines [51,52,53,54,57,61]. Ou et al. (2018) reported that silymarin supplementation to methionine–choline deficient (MCD) diet-induced NASH mice significantly diminishes levels of the pro-inflammatory cytokines tumor necrosis factor (TNF)-α, Interleukin (IL)-6, IL-1β and IL-12β [51]. Most of silymarin’s anti-inflammatory hepatoprotective effects have focused on cytokine release; however, Zhang et al. (2018) observed that silybin significantly impedes NLR family pyrin domain containing 3 (NLRP3) inflammasome activation in NAFLD by elevating NAD^+^ levels, which as a result preserves the effect of the NAD^+^-dependent α-tubulin deacetylase *sirtuin (**SIRT*)2 and restrains the activation of the acetylated α-tubulin promoted NLRP3 inflammasome, thus indicating the potential of silybin for targeting the NAD^+^/*SIRT2* pathway [54]. Apart from its anti-inflammatory action, several studies claimed that the immunomodulatory effect of silymarin and its bio-constituents could also play a remarkable role in hepatoprotection [14,62]. The anti-apoptotic and pro-apoptotic behavior of silymarin also considerably facilitates liver protection. Indeed, elevated oxidative stress and excessive discharge of pro-inflammatory cytokines can persuade apoptosis by stimulating the c-Jun NH2-terminal kinase (JNK) signaling pathway [63]. Kim et al. (2016) also showed that silymarin administration to stressed mice attenuates JNK activation and its associated apoptotic signaling by down-regulating the expression of Bid, Bax, caspase-3 and caspase-8 as well as poly adenosine diphosphate-ribose polymerase (PARP) cleavage [53].

Various animal and cell-based studies have revealed that silymarin and its bioactive constituents significantly impair the progression of initial liver fibrosis and its associated fibrogenetic mechanisms and induce a hepatoprotective behavior. In chronic liver injury, such as hepatitis C virus (HCV) infection, fibrosis and inflammation produce fibrous scarring through the activation of myofibroblasts in the liver, which consequently exudes extracellular matrix proteins. However, the augmentation of hepatic stellate cells (HSCs) and Kupffer cells is considered as the decisive episode in the production of hepatic fibrosis [14,64]. Experimental investigations have shown that silymarin impairs the proliferation of HSCs and prevents their translation into myofibroblasts, while also down-regulating gene expression of the extracellular matrix components required during fibrosis [57,65]. In a study by Clichici et al. (2015), silymarin was reported to decrease collagen and pro-collagen III by 30% after biliary obstruction in rats [57]. Activated HSCs also display an increase in expression of the monocyte chemoattractant protein-1 (MCP-1), which is an essential chemokine responsible for controlling monocyte/macrophage movement and permeation [66,67]. Mahli et al. (2015) have affirmed that silymarin treatment down-regulates MCP-1 and collagen 1 expression upon CCl_4_-induced hepatotoxicity in rats [46]. Silymarin also down-regulates the expression of α-smooth muscle actin (α-SMA), which directly triggers HSCs to activate myofibroblast-like cells [42,44,57]. Furthermore, the tissue inhibitor of metalloproteinases 1 (TIMP-1) also controls the alteration of the extracellular matrix in hepatic fibrosis via MMPs [68]. Silymarin significantly improves the level of MMP-2 and prevents fibrosis [44,61,69]. Notably, Chen et al. (2012) discovered that silymarin at a dose of 100 mg/kg significantly down-regulates transforming growth factor-beta 1(TGF-β), activator protein-1 (AP-1), α-SMA, MMP-2, MMP-13, collagen-α1 (COL-α1), TIMP-1, TIMP-2 and krueppel-like factor 6 (KLF6) expressions in a thioacetamide-induced hepatotoxicity rat model [44]. Furthermore, hepatocyte apoptosis prompts HSC activation and hepatic fibrosis; noteworthy treatment with silybin also reduces these changes [51].

Kupffer cells, on the contrary, induce fibrosis via the production of Kupffer cell-derived TGF-β1, which consequently activates myofibroblasts. Moreover, Kupffer cells also regulate the synthesis of MMP and TIMPs. Silymarin also significantly obstructs activation and function of Kupffer cells [57,70]. Likewise, several other pro-inflammatory cytokines, such as leptin and resistin, act as fibrogenic markers and stimulate fibrogenesis by activating portal fibroblasts, especially HSCs. Silymarin also down-regulates the expression of these fibrogenic markers and prevents hepatic fibrosis [40,42,51,52]. The anti-inflammatory ability of silymarin to impair nuclear factor-kappa B (NF-ĸB) also remarkably retards HSC proliferation [40].

Besides the aforementioned mechanisms, silymarin and its flavonolignans also induce hepatoprotective effects by anticipating liver regeneration and blocking toxic substances. Hepatocyte regeneration significantly recovers the liver from acute and chronic damage. It has been established that silymarin administration triggers hepatic regeneration by augmenting ribosomal RNA and RNA polymerase I synthesis, which consequently stimulates protein synthesis and repairs damaged liver cells [71,72].

### 2.2. Anti-Diabetic Activity

Increased diabetes mellitus (DM) and its associated complications are considered a global burden. DM is a progressive metabolic disorder characterized by chronic persistent hyperglycemia, insulin resistance and impaired insulin synthesis with elevated hepatic glucose outputs [73,74]. Silymarin and its constituents have been described for their potential hypoglycemic effects, and accumulating experimental and clinical evidence suggested that silymarin extensively trims down the blood glucose level and boosts insulin secretion (Table 2 and Figure 4) [75,76,77,78,79,80,81,82,83,84].

Results from streptozotocin (STZ)-induced diabetic rat models demonstrated that silymarin, when administered orally at a dose of 80 mg/kg for 21 days, remarkably reduces HbA1c levels and fasting blood sugar (FBS) levels [77]. Silymarin also imparts potential anti-diabetic activity by impeding gluconeogenesis and glucose-6-phosphatase (G6Pase) activity [85].

**Table 2 molecules-27-05327-t002:** Experimental anti-diabetic activity of silymarin.

Study Model	Dose/Concentration Used	Possible Target Site/Mechanism of Action	Reference
Obesity-induced insulin resistance model and HepG 2 cells	30 mg/kg/day p.o. for one month	Elevation in *SIRT1* expression Decrease in Akt and FOXO1 phosphorylation Increase in enzymatic activity of SIRT1	[86]
HFD model	30 mg/kg/day p.o. for one month	Decrease in insulin resistance Reduction in hepatic NADPH oxidase expression and NF-κB activity Decrease in GSH, CAT and SOD activity Reduction in IL-6, iNOS, NO and TNF-α levels	[75]
HFD-induced insulin resistance	30 and 60 mg/kg p.o.	Reduction in TNF-α, IL-1β and IL-6 levels Decrease in the levels of SGOT, SGPT, CH, TG and LDL Decrease in insulin resistance	[87]
HFD-induced insulin resistance model and HEK293T cells	40 μg/mL 50 µM	Reduction in FBS levels Inhibition of NF-κB signaling Activation of Farnesyl X receptor	[79]
Pancreatectomy model	200 mg/kg p.o	Increase in serum insulin levels Improvement in β cell proliferation Elevation in *Pdx1* and insulin gene expression	[88]
STZ-induced diabetes and INS1 cells	50 μg/mL 2.5–100 µM	Decrease in FBS and increase in insulin secretion Elevation in Bax and cleaved-caspase-3 protein levels Reduction in Bcl-2 and *pro-caspase-3* gene expression	[89]
STZ- and HFD-induced diabetes	100 and 300 mg/kg p.o.	Decrease in hepatic glucose production Increase in expression of the GLP-1 receptor in the duodenum	[90]
STZ-induced diabetes	60 and 120 mg/kg/day p.o. for 2 months	Down-regulation of *urotensin II* gene expression Reduction in FBS, CK-MB, LDH, MDA, CH, LDL and NO levels	[76]
STZ-induced diabetes	80 mg/kg p.o. for 21 days	Reduction in HbA1C levels Reduction in the levels of MDA, SGOT, SGPT, LDH and CK-MB in the heart Decrease in the levels of CH, TG and LDL An increase in Bcl-2 and decrease in Bax levels prevents apoptosis	[77]

Chronic hyperglycemia blights the mitochondrial respiratory chain and produces oxidative damage, resulting in the growth and development of DM and its associated complications. Treatment with silymarin significantly prevents oxidative damage by impeding lipid peroxidation, protein oxidation and reactive oxygen species (ROS) generation [75,91,92]. Qin et al. (2017) observed that silychristin A, one of the bioactive compounds of silymarin extracts, significantly protects ROS-induced apoptosis in INS 1 cells by elevating Bax and cleaved-caspase-3 protein levels and down-regulating *Bcl-2* and *pro-caspase-3* gene expressions [89]. Along with oxidative stress, inflammation is a key factor in diabetes progression and complications. Inflammatory cytokines have a decisive role in managing glucose homeostasis and insulin resistance. Any abnormal change in pro-inflammatory cytokines (IL-6 and TNF-α) could diminish insulin sensitivity and contribute to insulin resistance. In contrast, infiltrations of these cells can cause pancreatic β-cell failure [87]. Several studies have indicated that treatment with silymarin alleviates the inflammatory response by impeding the levels of NF-ĸB target genes [87,93]. It has been observed that silymarin also suppresses Interferon-γ (IFN-γ), TNF-α and IL-1β-induced nitric oxide (NO) generation, while also suppressing inducible nitric oxide synthase (iNOS) expression in pancreatic β-cells through modulating NF-κB activity and the extracellular signal-regulated kinase1/2 (ERK1/2) signaling pathway, which subsequently prevents pancreatic β-cell degradation [93].

An experiment by Xu et al. (2018) postulated that silybin decreases hepatic glucose production in STZ/high-fat diet (HFD) diabetics. Furthermore, it was observed that silybin also modulates the expression of the glucagon-like peptide (GLP)-1 receptor in the duodenum and activates neurons around the solitary tract, demonstrating the anti-diabetic potential of silymarin by eliciting the gut–brain–liver axis [90]. The Pdx1 transcription factor is believed to be directly involved in pancreatic growth and insulin gene expression, and the results from the study performed by Soto et al. (2014) revealed that silymarin elevates *Pdx1* and insulin gene expression in pancreatectomized rats along with improvements in β-cell proliferation. Furthermore, it was also reported that silymarin administration up-regulates *NKx6.1* gene expression, responsible for the differentiation, neogenesis and maintenance of β-pancreatic cells [94]. A current study by Feng et al. (2021) has demonstrated that silymarin administration elevates *SIRT1* expression in hepatocytes [86]. Furthermore, this study, also conducted on HepG2 cells in vitro, confirmed that silymarin binds to the SIRT1 enzyme and enhances its activity, thereby indicating the potent action of silymarin on insulin resistance and gluconeogenesis [86]. Silymarin and silybin also alleviate diabetes and other related metabolic syndromes by up-regulating Farnesyl X receptor signaling when studied in vitro using HEK293T cells [79].

Furthermore, it was also observed that silybin activates the insulin receptor substrate 1/phosphoinositide 3-kinase/protein kinase B (IRS-1/PI3K/Akt) pathway, which consequently elevates insulin-mediated glucose uptake and glucose transporter-4 (GLUT4) translocation [95]. Recent studies affirmed that estrogen receptors play an essential role in glucose metabolism and preserve islet β-cell functionality and viability. Silymarin treatment significantly up-regulates the expression of both estrogen receptors α and β and protects β-cells from the progression of DM [96,97,98].

Moreover, silymarin also attenuates DM-associated complications. It has been observed that silymarin also plays an imperative role in treating diabetic-induced neuropathy, nephropathy, cardiomyopathy, hepatopathy and delayed healing [99]. Recently, Rahimi et al. (2018) reported that silymarin down-regulates *urotensin II* gene expression, which is responsible for diabetic-associated cardiomyopathy by causing insulin resistance, inflammation and endothelial damage [76]. Furthermore, silymarin protects cardiomyocytes against DM-induced apoptosis by elevating Bcl-2 levels and down-regulating *Bax* expression [77,100]. Results from Meng et al. (2019) revealed that silymarin treatment also impedes the TGF-β1/Smad signaling pathway and improves cardiac fibrosis and collagen deposition in diabetic cardiomyopathy [101].

Silymarin also displays protective effects against diabetes-induced nephropathy. It significantly attenuates oxidative stress in the renal tissues by modulating the activity of various antioxidant enzymes [102,103]. Guzel et al. (2020) reported that silymarin at a 200 mg/kg dose significantly reduces caspase activity and the levels of malondialdehyde (MDA), NO and serum creatinine, while also possessing effective renal function that protects against vancomycin-induced nephropathy in rats [104]. Chen et al. (2021) demonstrated that chronic administration of silymarin down-regulates IL-6 and intercellular adhesion molecule-1 *(ICAM-1*) expressions and alleviates TGF-β/Smad and JAK2/STAT3/ SOCS1 pathways using an STZ-induced diabetic nephropathy model of rats with improvements in podoxin and nephrin levels [105]. Nevertheless, clinical studies also revealed that silymarin suppresses urinary TNF-α and MDA levels, indicating protective effects against diabetes-induced nephropathy [106]. Zhang et al. (2014) reported that oral administration of silybin to STZ- and HFD-induced diabetic rats for 22 days down-regulates the expression of retinal *ICAM-1*, contributing to the prevention of diabetic retinopathy [107].

### 2.3. Anti-Cancer Activity

Cancer is a group of diseases characterized by proliferation and differentiation in the growth of abnormal cells, invading normal tissues or organs and eventually spreading to all parts of the body. Cancer is a significant public health concern, and about 19.9 million new cancer cases were diagnosed in 2020 globally and are expected to increase to 28.4 million cases in 2040 [108,109]. Plants have been used as medicines by humanity for generations, and their considerably lower toxicity, ease of availability and high specificity towards targets, compared to synthetic chemotherapeutic agents, have stimulated much interest among researchers hoping to develop plant-based anticancer drugs. Moreover, recent advancements in isolation and identification of phytochemicals have also attracted heightened attention in relation to the application of herbal medicines as a prospective target for cancer management [110,111].

Excessive investigational studies revealed that silymarin possesses anti-cancer activity against almost all types of cancers, including colorectal cancer, bladder cancer, breast cancer, gastric cancer, prostate cancer, skin cancer, lung cancer, hepatocellular carcinoma, laryngeal carcinoma, glioblastoma and leukemia, as mentioned in Table 3 and Table 4 [7,112,113]. Silymarin and its bioactive constituents can curb the rise of different tumor cells, which is achieved by cell cycle arrest at the G1/S-phase, activation of cyclin-dependent kinase (CDK) inhibitors, reduction in anti-apoptotic gene product formation, obstruction of cell survival kinases and down-regulation of inflammatory transcription factors. Moreover, silymarin can also alter the expression of gene products associated with the proliferation of different tumor cells, their invasion, metastasis and angiogenesis [7,113]. The principal mechanisms of silymarin as an anti-cancer agent are mentioned in Figure 5. Nonetheless, silymarin and its bioactive constituents are also used in the prophylaxis of numerous anti-cancer therapy-induced side effects, such as the capecitabine-induced hand-foot syndrome [114], cisplatin-induced nephrotoxicity [115,116] and radiation-induced mucositis and dermatitis [117,118,119].

Numerous observations have reported that silymarin and its bioactive constituents attenuate cellular growth and proliferation by modulating mitogen-activated protein kinase (MAPK) signaling and inducing apoptosis in vitro [133,135,147,151,158,164]. MAPK is a key signaling pathway responsible for transferring extracellular stimuli to the nucleus. MAPK is further alienated into three subtypes: (i) ERK1/2, which imperatively regulates tumorigenesis, including cellular proliferation, division and viability; (ii) JNK; and (iii) p38, which significantly controls inflammation by modulating pro-inflammatory cytokine production and cell death [167]. Singh et al. (2002) were the first to report that silybin impedes cell proliferation and stimulates apoptosis in A431 cells by inhibiting MAPK/ERK1/2 activation and up-regulating stress-activated protein kinase/JNK1/2 (SAPK/JNK1/2) and p38 MAPK [151].

It was found from in vitro studies that silymarin considerably encourages apoptosis in both A2780s and PA-1 cells by elevating Bax and diminishing Bcl-2 protein expression, along with escalating caspase-9 and caspase-3 [149]. Similarly, Vaid et al. (2015) confirmed that silymarin encourages apoptosis of human melanoma cells via the down-regulation of anti-apoptotic proteins, mainly Bcl-2 and Bcl-xl, and the up-regulation of pro-apoptotic proteins, i.e., Bax along with activation of caspases [154]. Interestingly, Yang et al. (2013) demonstrated that silybin also down-regulates *survivin* expression, an essential part of the inhibitors of the apoptotic protein family, which subsequently induces apoptosis of Hep-2 cells [143].

A study by Won et al. (2018) endorsed that silymarin activates death receptor 5 to persuade apoptotic cell death in HSC-4, YD15 and Ca9.22 oral cancer cell lines [146]. Interestingly, it was also explored that silybin elevates death receptor 4/5 mRNAs and TNF-related apoptosis-inducing ligand (TRAIL) mRNA expression along with activation of caspase-9, thus indicating a dual mechanism of silymarin inducing potential anti-cancer activity through the augmentation of both the extrinsic and intrinsic apoptotic pathways [131,168]. In addition to inducing an apoptotic effect, silymarin also increases the levels of ceramides and modulates the secretion of micro RNA (miRNA) by down-regulating miR-92-3p and up-regulating miR223-3p and miR16-5p, which usually act as oncogenes and tumor suppressors in carcinogenesis [138]. Most importantly, Zhang et al. (2015) concluded that silybin prevents glioma cell proliferation and induces apoptosis via impeding *PI3K* and *Forkhead box M1 9* (*FoxM1*) expression, which further triggers the mitochondrial apoptotic pathway [169].

Furthermore, preventing cellular proliferation by disrupting typical cell cycle sequences and cellular divisions at various cell cycle stages is an important mechanistic approach to inducing anti-tumor activity. Cyclins, CDKs and cyclin-dependent kinase inhibitors (CDKIs) are critical cell cycle regulators which are overexpressed during cancer. Under normal physiological conditions, CDKs regulate the expression of genes involved in cell cycle transition. CDK2 with cyclin E and cyclin A, and CDK4 and 6 with cyclin D (CD), control G1/S cell cycle transition, whereas Cdc2 kinase with cyclin B regulates G2/M transition [159,170,171]. Silymarin and its bioactive flavonolignans significantly inhibit over-expression of these regulators and induce anti-carcinogenic activity [132,141,149,156]. It was observed from Western blot studies that silymarin impedes, in a dose-dependent manner, the gene expression of *CD1* and *CD2* along with a noteworthy diminution in protein expression of various CDKs in A375 cells. However, the levels of the CDK inhibitory proteins, Kip1/ p27 and Cip1/p21, were elevated after silymarin treatment [154]. Eo et al. (2015) reported that silymarin inhibits proliferation in HCT116 and SW480 cells by triggering the proteasomal degradation of CD1 at threonine-286 [132]. Moreover, Fan et al. (2014) demonstrated that silymarin dose-dependently inhibits the G1/S transition phase of the cell cycle in A2780s and PA-1 ovarian cancerous cells by altering *CDK2*, *p53*, *p21* and *p27* gene expressions [149]. Notably, a clinical study revealed that silymarin treatment in hepatocellular carcinoma patients also arrests the cell cycle pathway by down-regulating expression of the DNA topoisomerase 2-binding protein 1 (TOPBP1), the nucleolar and spindle-associated protein 1 (NUSAP1) and the cell division cycle-associated 3 protein (CDCA3), which are important for mitotic progression and regulation [172].

Despite inhibiting various CDK and MAPK signaling pathways, silymarin also significantly impedes various other pathways, such as the Notch pathway [141], the PI3K-PKB/Akt signaling pathway [120], the PP2Ac/ AKT Ser473/ mammalian target of rapamycin (mTOR) pathway [134], the Slit-2/Robo-1 signaling pathway [140,150] and the Wnt/β-catenin signaling pathway [129]. The PI3K/Akt pathway is an essential regulatory element responsible for cellular growth, proliferation and differentiation through activation of CDK-4 and CDK-2. Feng et al. (2016) reported that silybin impedes the PI3K/Akt/mTOR signaling pathway in U266 cells by down-regulating the protein expression of p-Akt, PI3K and p-mTOR [157]. Furthermore, it was shown that inhibition of PI3K/Akt pathways also restrains bladder cancer growth and progression in T24 and UM-UC-3 human bladder cancer cells [120]. The β-catenin-dependent Wnt signaling (Wnt/β-catenin signaling) pathway is also accountable for cellular proliferation, apoptosis and tissue homeostasis [173,174]. Numerous in vitro studies demonstrated silymarin anti-cancer activity via modulation of this signaling pathway [129,152]. Vaid et al. (2011) reported that silymarin treatment to A375 and Hs294t cells up-regulates expression of the glycogen synthase kinase-3β (GSK-3β) and the casein kinase 1α (CK1α), subsequently resulting in β-catenin phosphorylation and blockage of cellular migration and invasion [152]. Later on, Lu et al. (2012) observed that silymarin also modulates gene expression of the *lipoprotein receptor-related protein 6* (*LRP6*), a critical co-receptor for the Wnt/β-catenin signaling pathway [129]. Notch 1 signaling directly activates the cyclooxygenase-2 (COX-2)/Snail/E-cadherin pathway, which subsequently induces cancerous cell invasion and migration; however, silymarin’s ability to down-regulate Notch1 prevents this cellular invasion and results in decreased cancer growth [141,175].

Matrix metalloproteinases (MMPs) degrade extracellular matrices and are accountable for metastasis and migration of cancerous cells in carcinogenesis. Silybin was reported to significantly impede *MMP-2* gene expression by modulating Janus kinase-2/signal transducers and activators of the transcription-3 (Jak2/STAT3) pathway [126]. MMP plays a pivotal role in cancer metastasis, together with AP-1, through stimulation of the epithelial to mesenchymal transition of tumor cells. Down-regulating AP-1 may also serve as a possible remedial approach for cancer treatment [176]. It was reported that silybin acts as a potential anti-metastasis agent by remarkably suppressing cellular invasion by impeding *AP-1 dependent MMP-9* gene expression in MCF-7 human breast carcinoma cells [127]. Recently, Si et al. (2020) found that silybin treatment elevates mitochondrial fusion through an increase in the expression of mitochondrial fusion-associated proteins (optic atrophy 1, mitofusin 1 and mitofusin 2) and down-regulation of the expression of the mitochondrial fission-associated protein (dynamin-related protein 1 (DRP1)) in MDA-MB-231 breast cancer cells, which consequently impedes cellular migration [124]. Furthermore, it was observed that silybin treatment also abolishes activation of the NLRP3 inflammasome through repression of ROS generation, resulting in reduced tumor cell migration and invasion [124]. The C-X-C chemokine receptor type in cancer cells is accountable for tumor growth, proliferation and angiogenesis [177]. Kacar et al. (2020) demonstrated that silymarin activates SLIT2 protein and suppresses C-X-C chemokine receptor type 4 expressions in DU-155 cells, thereby reducing tumor proliferation and metastasis in prostate cancer [150].

The vascular endothelial growth factor (VEGF) displays a significant role in tumor-induced angiogenesis and can be used as a hopeful target for anti-cancer therapy. Western blotting analysis demonstrated that a 500 mg/kg silymarin treatment given to mice remarkably down-regulated VEGF protein expression in the A375 tumor xenografts model [154]. Furthermore, silymarin administration was also reported to reduce CD31 expression levels, contributing to the development of new vasculature. Most interestingly, Khan et al. (2014) revealed that 9 mg/mouse topical applications of silybin significantly repress tumorigenesis and oxidative stress by down-regulating iNOS, NO, TNF-α, IL-6, IL-1β, COX-2 and NF-κB [153]. Nowadays, drug resistance to cancer therapies is a major problem in combating this disease, and various studies affirm that feedback activation of STAT-3 is majorly responsible for mediating drug resistance. However, considerable evidence suggests that silymarin can reverse STAT-3-associated cancer drug resistance by down-regulating its gene expression [178].

### 2.4. Neuroprotective Activity

Silymarin and its flavonolignans are also involved in the treatment of various neurogenerative diseases and attenuation of the neurodegenerative alteration developed after cerebral ischemia. The main neuroprotective mechanisms of silymarin are illustrated in Figure 6. Kittur et al. (2002) demonstrated silymarin’s neurotrophic and neuroprotective effects via augmentation of nerve growth factor (NGF)-mediated neurite outgrowth in PC-12 neural cells and protection against oxidative stress-triggered cell death in rat hippocampal neurons [179]. Nitrosative stress, oxidative stress and neuroinflammation are indeed imperative pathological traits of various neurodegenerative disorders, including Alzheimer’s and Parkinson’s diseases [112,180]. Silymarin treatment significantly suppresses neuronal oxidative stress and neuroinflammation by increasing numerous exogenous enzymatic activities and attenuating inflammatory responses, respectively, as discussed in Table 5 [181,182,183,184,185,186,187]. Recently, it was also observed that silymarin intake prevents glutamate release via inhibition of voltage-dependent Ca^2+^ channels and the ERK1/2 pathway in a rat model of kainic acid-mediated excitotoxicity [188].

Parkinson’s disease is the most common neurodegenerative disorder and is characterized by progressive loss of dopaminergic neurons with pervasive intracellular aggregation of the α-synuclein protein in the substantia nigra pars compacta causing irregularities in motor behavior [189,190]. Silymarin displays potential anti-Parkinson activity by upholding striatal dopamine levels via inhibition of apoptosis and protection of dopamine neurons in the substantia nigra [191,192]. Srivastava et al. (2017) reported that silymarin treatment reduces α-synuclein protein levels through modification in mRNA expression of various α-synuclein suppressive genes in a *Caenorhabditis elegans* transgenic model [193]. Moreover, silymarin’s ability to hamper monoamine oxidase-B (MAO-B) enzymatic activity adds to the neuroprotective mechanisms of silymarin which counteract the loss of dopamine in parkinsonism [194,195]. Most importantly, Wang et al. (2002) observed that silymarin remarkably prevents microglia (glial cell) activation by inhibiting NF-κB signaling, which furthermore impairs dopamine neuron damage and likely represents a possible mechanism for its neuroprotective activity [196]. Silymarin treatment also augments the expression of the lysosome-associated membrane protein-2 (LAMP-2A) protein and reduces p-AMPK-mediated Ulk1-dependent macroautophagy in the MPTP-induced Parkinson model [197].

Alzheimer’s disease is a progressive neurodegenerative disease exemplified by progressive amyloid-beta (Aβ) peptide aggregation, synapse and nerve destruction and acetylcholine deficiency, causing memory impairment along with functional and behavioral alterations [112,198]. Inhibition of Aβ aggregation and its associated oxidative stress, as well as acetylcholinesterase (AChE) activity, is a prospective beneficial approach that can slow or diminish the development of Alzheimer’s disease [112]. Various in vivo and in vitro experimental studies affirm that silymarin and its bioactive constituents display anti-amyloidogenic activities and are capable of impairing Aβ fibril formation and aggregation along with inhibition of AChE enzymatic activity [198,199,200,201,202]. The amyloid precursor protein (APP) is liable for the formation of Aβ peptides through its sequential proteolytic cleavages and has a central role in the growth of Alzheimer’s disease [203]. Yaghmaei et al. (2014) demonstrated that silymarin remarkably down-regulates APP gene expression [201]. Furthermore, it was also found that silymarin reduces Aβ-protein fibril formation in an APP transgenic animal model, indicating the role of silymarin in the inhibition of APP expression [200,202]. Oxidative stress is also extremely accountable for Aβ toxicity and displays a critical role in the pathophysiology of Alzheimer’s disease. Silymarin was found to have a protective effect against Aβ-mediated oxidative stress [199,204,205]. Zhou et al. (2016) reported that isosilybin attenuates oxidative stress-induced Aβ peptide formation in hippocampal neuronal cells, possibly via augmentation of the NRF2/ARE signaling pathways [205]. Recently, it was also found that estrogen receptors play a significant role in protection against Aβ-induced toxicity, and treatment with silymarin notably modulates estrogen receptor activity and its related signaling pathways, such as MAPK and PI3K-Akt [206,207].

**Table 5 molecules-27-05327-t005:** Experimental studies demonstrating the activity of silymarin on CNS.

Pharmacological Activity	Study Model	Dose/Concentration Used	Possible Target Site/Mechanism of Action	References
Neuroprotective	Lipopolysaccharide (LPS)-induced neuroinflammatory impairment	25–100 mg/kg	Over-expression of *BDNF* and *TrkB* genes Reduction in *IL1β, NF-ĸB* and *TNF-α* expression	[183]
	Docetaxel-induced central and peripheral neurotoxicities	25 and 50 mg/kg	Down-regulation of *NF-ĸB*, *TNF-α*, *Bax* and *JNK* expression Up-regulation of Nrf2, *Bcl-2*, *CREB* and *HO-1* expression Reduction in the levels of GSH and SOD	[208]
	STZ-induced diabetic neuropathy	30, 60 mg/kg/day p.o.	Reduction in the levels of SGPT, SGOT, INFγ, IL-1β, IL-6, TNF-α, TBRAS and endogenous antioxidant enzymes Increase in the levels of TP and albumin	[182]
	—	30–300 g/mL	Inhibition of MAO-B and activation of Na^+^/K^+^-ATPase	[195]
	Manganese-induced neurotoxicity	100 mg/kg/day i.p.	Reduction in AOPP, PCO, TBARS and NO levels in the cerebral cortex Elevation in antioxidant enzyme activities	[209]
	Acrylamide-induced cerebellar damage	160 mg/kg	Elevation in 5-HT and dopamine levels Reduction in MDA levels Increase in CAT and SOD levels	[210]
	Ischemic surgery	200 mg/kg	Postponement of neuronal cell death	[211]
	Kainic acid (KA)-induced excitotoxicity	50–100 mg/kg	Suppression of synaptosomal glutamate release Inhibition of ERK1/2 activity Blockage of voltage-gated Ca^2+^ channels	[188]
	Middle cerebral artery occlusion	—	Amplification of pAkt, *HIF-1α*, *pmTOR* and *Bcl-2* expression Down-regulation of *Bax* and NF-κB expression Activation of the Akt/mTOR signaling pathway	[212]
Anti-Alzheimer	Aβ_1–42_-induced Alzheimer’s	70 and 140 mg/kg p.o. for 4 weeks	Inhibition of amyloid plaque formation Down-regulation of APP gene expression	[201]
	APP transgenic mice and PC12 cells	0–100 µM	Decrease in A β-protein fibril formation Improvement in behavioral abnormalities	[202]
	APP/PS1 transgenic mice	2–200 mg/kg/day	Inhibition of AChE activity Reduction in plaque formation	[200]
	Scopolamine-induced dementia	200–800 mg/kg p.o. for 2 weeks	Diminution in AChE activity and MDA level Restoration of dopamine and GABA activity Down-regulation of GFAP and NF-κB protein expression	[199]
	Aβ_1–42_-induced Alzheimer	25–100 mg/kg	Modulation of estrogen receptor α and β expression Inhibition of MAPK and PI3K-Akt pathways	[206]
	Aluminum chloride (AlCl_3_)-induced Alzheimer’s	34 mg	Suppression of AChE activity	[198]
	Aβ_25–35_-induced Alzheimer’s	25–100 mg/kg	Elevation in autophagy level Decrease in COX-2, NF-κB and iNOS expression Elevation in IL-4 levels	[181]
	Aβ25–35-induce oxidative stress damage in HT-22 cells	—	Activation of the NRF2/ARE pathway	[205]
Anti-Parkinson	*Caenorhabditis elegans* transgenic model	24.12 µg/mL	Decrease in α-synuclein protein levels Alteration in mRNA expression of α-synuclein suppressive genes Elevation in dopamine levels	[193]
	MPTP-induced parkinsonism	40 mg/kg i.p. for 2 weeks	Decrease in beclin-1, α-synuclein, sequestosome, p-Ulk1 and p-AMPK levels Elevation in DA, LAMP-2 and p-mTOR levels	[197]
	6-OHDA-induced neurodegeneration and parkinsonism	100 and 200 mg/kg i.p.	Inhibition of TBARS formation Protection of substantia nigra	[192]
	6-OHDA-induced neurodegeneration and parkinsonism	100, 200 and 300 mg/kg, i.p. for 5 days	Improvement in motor coordination Elevation in MDA levels Reduction in CSF level of IL-1β	[185]
	MPTP-induced parkinsonism	20–400 mg/kg, i.p.	Preservation of dopamine level and dopamine neurons in the substantia nigra Reduction in apoptotic cells	[191]
Anti-depression	—	5–200 mg/kg	Elevation of NO levels	[213]
	Olfactory bulbectomized (OBX) technique	100–200 mg/kg	Improvement in BDNF expression Reduction in MDA, IL-6, TNF-α levels and oxidative stress Elevation of dopamine levels	[184]
	Reserpine-induced depression	0–400 mg/kg	Up-regulation of BDNF and TrkB expression Improvement in neuronal stem cell proliferation Enhancement in p-ERK and p-CREB levels	[214]
	Aβ_1–42_-induced Alzheimer’s	25–100 mg/kg	Enhancement in *BDNF* and *TrkB* expression	[215]

Apart from its effect on various neurodegenerative disorders, silymarin and its constituents also help manage depression. Depression is a mood disorder characterized by an importunate feeling of unhappiness and loss of interest and is directly allied with brain-derived neurotrophic factors (BDNF) and neuroinflammation [112]. Indeed, BDNF is a neurotrophin neuronal growth, function and survival factor, and impairment in BDNF/ tropomyosin receptor kinase B (TrkB) signaling is considered a potential underlying factor for depression [216,217]. Several experimental studies demonstrated that silymarin and its associated constituents elevate BDNF levels, and also impair inflammatory responses and oxidative stress, via amplification of the BDNF/TrkB pathway [183,184,214,215]. In addition to this, a study by Khoshnoodi et al. (2015) has revealed that silymarin’s capability to elevate NO levels modulates the effect of various neurotransmitters, such as serotonin, norepinephrine and dopamine, which consequently induces antidepressant-like effects [213].

Silymarin was also found to ameliorate cerebral ischemia and cerebral stroke by interrupting neurodegenerative progression and inhibiting neuronal cell death in several ischemia models [211,218,219]. Wang et al. (2012) observed that silybin treatment before permanent middle cerebral artery occlusion significantly activates the Akt/mTOR signaling pathway and induces protection against ischemic stroke. Furthermore, it was affirmed that silybin also up-regulates *hypoxia-inducible factor 1α* (*HIF-1α*) and *Bcl-2* expression and down-regulates *Bax* and *NF-κB* expression in ischemic brain tissue after stroke [212].

### 2.5. Cardioprotective and Anti-Hypertensive Activity

Cardiovascular diseases are currently the foremost source of fatality in aged adults and are usually allied with ischemic injury [112]. However, their increasing prevalence has warranted the attention of researcher who are now seeking to generate newer solutions to curb this problem and promote cardiac health. In this regard, various in vitro and in vivo studies have been carried out that describe the cardioprotective effects of silymarin, which directly plummets the levels of MDA, *lactate dehydrogenase* (LDH), troponin C and creatine kinase-MB (CK-MB), which are important cardiac biomarkers [77,220,221]. Moreover, silymarin treatment also diminishes myocardial oxidative stress by elevating catalase (CAT) and superoxide dismutase (SOD) activities as well as GSH content in the heart [222]. Apart from reducing the level of cardiac enzymes, silymarin also protects the heart by arresting cardiomyocyte apoptosis. Taghiabadi et al. reported that silymarin decreases the Bax/Bcl-2 ratio, cytosolic cytochrome c content and cleaved caspase-3 levels of the heart [222]. Later on, a study by Alabdan (2015) also showed that pre-treatment with 80 mg/kg of silymarin orally could prevent hyperglycemia and myocardial injury in STZ-treated diabetic rats [77].

Notably, silymarin intake in CCl_4_-intoxicated rats markedly decreases the level of VEGF, which is an important angiogenic biomarker. Furthermore, it was observed that silymarin ameliorates the level of inflammatory and immunological biomarkers, such as TNF-α, InFγ, IL-6 and C-reactive protein (CRP), and marks the anti-inflammatory potency of silymarin in protection against myocardial injury [220]. Gabrielová et al. (2015) observed that 2,3-dehydrosilybin, a constituent of silymarin, remarkably increases *luciferase* gene expression and intracellular cAMP levels while also inducing inhibition of the phosphodiesterase enzyme in isolated neonatal rat cardiomyocytes [223].

Besides this, silymarin also plays a critical role in reducing elevated atrial blood pressure (B.P.). Silymarin significantly reduces systolic B.P., basal arterial B.P. and heart rate in both the DOCA salt and fructose-induced hypertension model. It was also observed that silymarin treatment in hypertensive rats decreases urinary K^+^ excretion and tribarbituric acid reactive substance (TBARs) levels along with an augmentation in urinary Na^+^ excretion and endogenous antioxidant enzyme levels, as mentioned in Table 6 [224,225].

### 2.6. Anti-Viral Activity

Viral infections are considered a menace to public health and enhance the global socioeconomic burden. Currently, the significant increase of viruses as human pathogens and the rise of large-scale epidemic outbreaks have highlighted a demand for new anti-viral drugs. Silymarin depicts inhibitory action against many viruses in various cell lines and in vivo studies by targeting manifold steps of the viral life cycle, as discussed in Table 7. Silymarin and silybin directly hinder infection of the hepatitis C virus (HCV) in cell cultures by blocking viral entry, viral fusion, viral RNA and viral protein synthesis along with polymerase activity and virus transmission [56,226,227,228,229]. Silymarin may reveal an advantageous relationship with viral hepatitis through its inhibitory potential on inflammatory processes and the cytotoxic cascade of events triggered by viral replication [230,231]. Polyak et al. (2007) reported that, besides impairing HCV RNA and protein expression to prevent viral replication, silymarin also inhibits TNF-α secretion and NF-ĸB-dependent transcription, thus indicating the dual anti-viral and anti-inflammatory activity of this extract [232]. Furthermore, data suggest that certain bioactive components of silymarin significantly inhibit HCV infection via modulating the Jak-STAT pathway. Moreover, an in vivo study using the uPA^+/+^/SCID^+/+^ chimeric mice model by DebRoy et al. (2016) concluded that intravenous (iv) monotherapy with silybin significantly inhibits HCV production and prevents the production of various transcription regulators and inflammation-related cytokines [233].

In another work, Blaising et al. (2013) demonstrated that both silybins A and B significantly block HCV entry into the cell through clathrin-coated pits and vesicles by slowing HCV endosomal trafficking and inhibiting clathrin-mediated endocytosis, thereby preventing HVC infection [234]. Consistent with the above results, it was also demonstrated that oral concomitant treatment of silybin–vitamin E–phospholipid complex pills with PEG-IFN and ribavirin in chronic hepatitis patients for 12 months significantly lowers the viral load [235,236]. Furthermore, iv infusion of silymarin also displayed significant anti-HCV activity in clinical studies [229,237]. HCV-associated liver cirrhosis and liver carcinoma are common after liver transplantation. Clinical data also reported that silymarin and its bioactive constituents prevent HCV reoccurrence after liver transplantation [238,239].

As in HCV infection, silybin also hinders clathrin-mediated endocytosis in the hepatitis B virus (HBV), thus inhibiting HBV entry into cells [240]. Apart from its anti-hepatitis potential, silymarin possesses an anti-viral activity against other viral infections, such as the severe acute respiratory syndrome coronavirus 2 (SARS-CoV-2) [241], the influenza virus [242,243], the dengue virus [244,245], the mayaro virus [246,247], the enterovirus 71 [248], the chikungunya virus [249,250], the herpes virus [251] and the human immunodeficiency virus (HIV) [252,253], by similarly inhibiting viral entry, viral fusion, viral RNA and viral protein synthesis. Some studies revealed that silymarin impairs influenza replication dose-dependently via inhibition of mRNA synthesis [242,243]. Interestingly, the 23-(*S*)-2-amino-3-phenylpropanoyl derivative of silybin also blocks viral replication by impeding formation of the Atg5-Atg12/Atg16L complex and amplifying infection-induced autophagy. In addition, this silybin derivative also improves MAPK/ERK/p38 and IkB kinase (IKK) signaling pathways [243].

**Table 6 molecules-27-05327-t006:** Experimental studies demonstrating the effect of silymarin on the cardiovascular system.

Study Model	Dose/Concentration Used	Finding/Possible Mechanism of Action	Reference
CCl_4_-induced cardiomyopathy	200 mg/kg/day p.o. for 21 days	Reduction in CK-MB, Troponin-T, INFγ, IL-6, TNF-α, CRP and VEGF levels by silymarin	[220]
Ischemia reperfusion-induced myocardial infarction	100–500 mg/kg p.o. for one week	Significant decrease in the levels of MDA, SGOT, SGPT, LDH, CK-MB, CK and endogenous antioxidant enzymes.	[221]
Acrolein-induced cardio toxicity	25–100 mg/kg p.o.	Significant decrease in the levels of MDA, troponin T, CK-MB and endogenous antioxidant enzymes Inhibition of apoptosis by a reduction in the Bax/Bcl-2 ratio, cytosolic cytochrome c content and cleaved caspase-3 levels in heart	[222]
Isoproterenol-treated rat cardiac myocytes	0–0.7 mM	Reduction in SOD, LDH and MDA levels Up-regulation of *SIRT1* and *Bcl-2*	[254]
Perfused adult rat heart model and H9c2 cells	0.01–10 µM	Elevation in *luciferase* gene expression and intracellular cAMP levels Inhibition of phosphodiesterase enzyme	[223]
Doxorubicin-induced cardio toxicity and hepatotoxicity	60 mg/kg p.o. for 12 days	Decrease in the level of SGOT, SGPT, LDH, CK-MB and endogenous antioxidant enzymes	[255]
DOCA salt-induced hypertension	300 mg/kg and 500 mg/kg, p. o. for 4 weeks	Decrease in systolic B.P., basal arterial B.P. and heart rate Elevation in urinary Na^+^ excretion and endogenous antioxidant enzymes levels Reduction in urinary K^+^ excretion and TBARS levels	[224]
Fructose-induced hypertension	300 mg/kg and 500 mg/kg, p. o. for 6 weeks	Decrease in systolic B.P., basal arterial B.P. and heart rate Elevation in endogenous antioxidant enzymes levels Reduction in TBARS levels	[225]

**Table 7 molecules-27-05327-t007:** Experimental anti-viral activity of silymarin.

Type of Viral Infection	Study Model	Dose/Concentration Used	Possible Target Site/Mechanism of Action	Reference
Chikungunya virus	Vero and BHK-21 cells	50 μg/mL	Reduction in viral replication efficacy Down-regulation of the production of viral proteins involved in the replication	[249]
HCV	Huh 7 cells	—	Inhibition of NS5B RNA-dependent RNA polymerase activity Inhibition of HCV and JFH1 replication	[226]
	HepG2 and Huh 7 cells	—	Reduction in JFH-1RNA and HCV RNA production Inhibition of MTP-dependent apoB secretion	[227]
	Huh 7 and PBM cells	20–200 μg/mL	Stimulation of the Jak-Stat pathway Inhibition of TNF-α secretion and NF-ĸB-dependent transcription Inhibition of HCV RNA and protein expression	[232]
	Huh7.5 cells	10.4–150 µM	Inhibition of the clathrin-dependent pathway by inhibiting HCV endosomal trafficking and clathrin-mediated endocytosis	[234]
	—	1–1000 µM	Inhibition of the HCV NS4B protein	[256]
	uPA^+/+^/SCID^+/+^ chimeric mice model	61.5, 265 and 469 mg/kg i.v. for 14 days	Decline in HCV production Elevation in anti-inflammatory and anti-proliferative gene expression	[233]
Influenza virus	MDCK cells	100 μg/mL	Inhibition of mRNA synthesis	[242]
	MDCK, A549 and Vero cells Viral infection of BALB/c mice	25 mg/kg/day	Activation of MAPK/ERK/p38 and IKK signaling pathways Inhibition of viral replication and formation of the Atg5-Atg12/Atg16L complex Enhancement in infection-induced autophagy	[243]
HIV	PBMC and CEM-T4 cells	50–500 µM	Inhibition of T cell mitochondrial respiration and glycolysis Inhibition of HIV entry	[253]
	TZM-bl, PBMC and CEM cells	40–324 µM	Inhibition of viral replication Reduction in CD4+, CD8+ and CD19+ T cell proliferation Blockage of activation markers on CD4+ T cells	[252]
Mayaro virus	HepG2 cells	3.125–1400 μg/mL	Decrease in MDA levels and ROS formation	[246]
HBV	HepG2-NTCP-C4 cells	0–200 μM	Inhibition of clathrin-mediated endocytosis and reduction in transferrin uptake	[240]
Herpes virus	Vero cell	0–125 μg/mL	It reduced the IC_50_ value to 100 μg/mL	[251]
SARS-CoV-2	Human umbilical vein endothelial cells	5–25 µM	Down-regulation of *TNF-α*, *IL-6, MCP-1* and *PAI-1* gene expressions	[241]
	—	1–100 µM	Inhibition of M^pro^ (main protease)	[257]

Importantly, silymarin exerts its anti-HIV activity by modulating various cellular functions that are responsible for T cell activation and proliferation. Silymarin significantly reduces CD4+, CD8+ and CD19+ T cell proliferation and blocks activation markers such as HLA-DR, CD38, CCR and Ki67 on CD4+ T cells, which consequently results in fewer CD4+ T cells expressing the HIV co-receptors (the C-X-C chemokine receptor 4 and the C-C chemokine receptor 5) [252]. Furthermore, another study by McClure et al. (2014) demonstrated that silymarin disturbs T cell metabolism by impairing mitochondrial respiration and glycolysis, which is useful in combating HIV infection while simultaneously blocking viral replication [253]. Moreover, based on the dual potential of silymarin to prevent both HCV and HIV viral replication, many clinical trials have been performed using this bioactive drug for HCV/HIC coinfected patients [258,259,260]. Recently, in line with the above results, many clinical trials are underway to study and understand the potential prophylactic or therapeutic activity of silymarin and its bioactive flavonolignans against COVID-19 [13,241,257,261,262].

### 2.7. Photoprotection and Dermal Applications

Skin is the outermost protective organ of the human body, defending it from oxidants and several exogenous pollutants [263,264]. Ultraviolet (UV) radiation-induced ROS generation modulates several cellular pathways and the expression of various inflammatory cytokines that, consequently, alter epidermal cellular activity. Moreover, elevated collagenase, elastase and hyaluronidase enzymatic activities also result in photoaging [265]. Accumulating data divulged that silymarin and its bioactive compounds have a pivotal role in skincare and can be used to treat various skin disorders such as melasma, photo-aging, rosacea, atopic dermatitis, psoriasis, acne and radiodermatitis, as mentioned in Table 8 and Table 9.

DNA mutation and the generation of cyclobutane pyrimidine dimers (CPDs) are the common mechanisms that explain the photo-protective activity of silymarin, though it was recently reported that silymarin also repairs UVB-induced DNA damage by augmenting the expression of various nucleotide excision repair (*NER*) genes [266,267]. In XPA-deficient and XPA-proficient mice models, silymarin treatment remarkably reduces sunburn/apoptotic cell counts in NER-proficient mice. However, no significant change was observed in their wild-type counterparts, thus confirming the role of *NPR* gene expression in silymarin-mediated photoprotection [267]. Adjacent to the NER pathway, the p53 signaling pathway plays a crucial role in photoprotection. Several experimental studies have established that up-regulation of p53-mediated growth arrest and *DNA damage-inducible protein α* (*GADD45α*) expression is a decisive mechanism by which silymarin protects against UVB-induced photo damage [266,268,269]. GADD45α is indeed a vital transcription factor that is mediated by p53 and which regulates apoptosis, cellular proliferation and DNA damage repair [270,271]. Additionally, Rigby et al. (2017) determined the role of p53 in UVB photoprotection using p53 heterozygous (*p53*^+^/^−^) and *p53* knockout (*p53*^−^/^−^) mice. Results revealed that silybin treatment considerably reduces UVB-induced lesions in *p53*^+^/^+^ compared with p53 deficient mice, affirming the role of silymarin in photoprotection via p53 activation [269]. Moreover, an earlier study by Gu et al. (2005) reported that silymarin hampers JNK1/2, ERK1/2, MAPK/p38 and AKT signaling pathways during UV-induced mitogenesis and prevents skin from light damage [272]. Silymarin and its major constituents can also induce photoprotection by preventing DNA single-strand break (SSB) formation and ROS generation along with a decrease in the levels of HSP70, MMP-1 proteins and *caspase-3* activation [273,274].

The anti-inflammatory, antioxidant and anti-apoptotic potential of silymarin also has a significant role in photoprotection and impairing UV-induced oxidative stress [275,276,277,278]. Juráňová et al. (2019) showed that silymarin attenuates skin inflammation by activating NF-κB and AP-1 through up-regulation of IL-8 mRNA, which consequently protects from UVB-induced light damage [279]. Silymarin also significantly prevents UV-induced apoptosis by impairing caspase-3 and 8 activity [280,281].

**Table 8 molecules-27-05327-t008:** Experimental studies demonstrating the dermal applications of silymarin.

Pharmacological Activity	Study Model	Dose/Concentration Used	Possible Target Site/Mechanism of Action	Reference
Photo protective	UV exposure	0.1–0.2 mg/mL/kg topically	No skin irregularity, erythema, hyperpigmentation or edema were observed	[282]
	UV exposure	—	Impedes SSB production and ROS generation Decreases HSP70, MMP-1 and caspase-3 level Increases HO-1 level	[274]
	UV exposure in HaCaT cells	75 μm	Decrease in *caspase-3* activation and ROS levels Up-regulation of CHOP protein expression	[281]
	UVB-induced skin damage in human dermal fibroblasts	1.6–100 μM	Decrease in *caspase-3* activation and ROS levels	[273]
	XPA-deficient mice, XPA-deficient and XPA-proficient human fibroblasts and normal human epidermal keratinocytes	10 and 20 µg/mL	Reduction in apoptotic cell count Up-regulation of *NPR* gene expression	[267]
	JB6 cells and mouse skin	100 μm	Impediment of cell cycle progression Up-regulation of *GADD45α* gene expression	[268]
	Human dermal fibroblasts	100 μm	Elevation in *p53* and *GADD45α* gene expressions	[266]
	SKH-1 hairless mouse	9 mg topically	Activation of the *p53* pathway	[269]
	SKH-1 hairless mice skin	9 mg topically	Reduction in MAPK and AKT signaling pathways Elevation in the p53 signaling pathway	[272]
Anti-alopecia	Human dermal papilla cells	0–200 μM	Elevation in luciferase enzymatic activity Activation of the AKT and Wnt/β-catenin signaling pathway	[283]
Wound healing	Human fibroblast cells	4.5–36 µg/mL	Down-regulation of COX-2 mRNA expression	[284]
	Rat wound model with full-thickness excision	2% ointment containing 500 mg silymarin	Reduction in redness, swelling and exudation Decrease in MDA levels Elevation of NO synthase expression and estradiol levels	[285]
	Rat wound model full-thickness cutaneous defect	6–12 mg/mL	Decrease in lymphocyte and macrophage counts Elevation in fibrocytes count, college fibers and fibroblasts Improvement in tensile strength	[286]
	Normal human dermal fibroblasts	—	Up-regulation of IL-8 mRNA Activation of NF-κB and AP-1 Reduction in IL-6 and IL-8 release	[279]
Anti-aging	—	0.01–2.5 g/L	Inhibition of collagenase and elastase enzyme activities	[287]

**Table 9 molecules-27-05327-t009:** Clinical evidence published in the previous 10 years depicting various pharmacological activities of silymarin.

Disease	No. of Patients	Dose; Duration	Add on Therapy	Study Outcomes	Reference
Diabetes	40	140 mg tid p.o.; 90 days	—	Decrease in FBS, HbA1c, MDA, CH, TG and LDL	[80]
	40	140 mg tid p.o.; 45 days	—	Reduction in FBS, SOD, MDA and hs CRP levels	[288]
	40	420 mg tid p.o.; 45 days	—	Reduction in HOMA-IR, insulin, LDL CH and TG levels Increase in HDL levels	[289]
	85 (diagnosed with Type 1 diabetes)	105 mg bid p.o.; 6 months	*Berberis aristata* 588 mg	Reduction in FBS, HbA1c, LDL CH and TG levels Increase in HDL levels	[290]
	69	1000 mg/day p.o.	Berberine 210 mg/day	Decrease in FBS, HbA1c, SGPT, SGOT, CH, TG and LDL levels	[291]
Dyslipidemia	139	105 mg bid p.o.; 6 months	*Berberis aristata* 500 mg and Monacolin K 10 mg	Reduction in FBS, LDL CH and TG levels Inhibition of TNF-α and IL-6 release	[38]
	137	105 mg bid p.o.; 6 months	*Berberis aristata* 588 mg	Reduction in FBS, insulin and HOMA-index levels Improvement in lipid profile	[37]
	105	105 mg bid p.o.; 3 months	*Berberis aristata* 588 mg	Reduction in retinol-binding protein-4 and resistin levels Increase in adiponectin levels	[36]
Melasma (skin disorder)	96	7 and 14 mg/mL cream bid topically; 4 weeks	—	Melasma area and severity index (MASI) reached zero after 4 weeks	[282]
Acne	20	1% seed oil cream bid topically	—	Reduction in facial wrinkles and improvement of skin tone	[292]
	56	—	*N*-acetylcysteine and Selenium	Reduction in MDA and IL-8 levels Decrease in the number of inflammatory lesions	[293]
Hepatocellular carcinoma	40	—	—	Reduction in CDCA3, TOPBP1 and NUSAP1 levels	[172]
Cisplatin-induced nephrotoxicity	60	140 mg bid p.o.; 7 days	—	Decrease in BUN and creatinine levels	[116]
	86	140 mg tid p.o.; 21 days	—	Decrease in serum creatinine levels	[115]
Capecitabine-induced hand-foot syndrome	40 (diagnosed with G.I.T. cancer	1% gel bid topically; 9 weeks	Capecitabine	Minimizes the severity of the syndrome and impairs its incidence	[114]
Radiotherapy-induced mucositis	27 (Diagnosed with head and neck cancer)	420 mg/ day p.o.; 6 weeks	—	Significant delay in mucositis growth and progression	[117]
Radiation-induced dermatitis	40 (Diagnosed with breast cancer)	1% gel bid topically; 5 weeks	—	Significant delay in dermatitis growth and progression	[118]
NAFLD	81	280 mg bid p.o.; 90 days	Vitamin C 120 mg, Vitamin E 40 mg, Coenzyme Q10 20 mg and Selenomethionine 83 µg	Reduction in the levels of SGPT, SGOT, ALP and γ-GT	[294]
	66	140 mg/day p.o.	—	Decrease in SGPT, SGOT and lipid profile levels Reduction in FBS, serum insulin levels and HOMA index	[295]
	36	540 mg bid. p.o.; 3 months	Vitamin E	Decrease in γ-GT and fibrosis scores	[296]
	30	188 mg p.o.; 6 months	Vitamin E and Phospholipids	Reduction in fatty liver index levels	[297]
	179	94 mg bid. p.o.; 12 months	Phosphatidylcholine 194 mg and Vitamin E 89 mg	Improvement in SGPT, SGOT, γ-GT, TGF-β and MMP-2 levels	[69]
	150	303 mg bid. p.o.; 6 months	Vitamin D 10 mg and Vitamin E 15 mg	Reduction in the levels of HOMA-IR, CH, TG, IL-18, IL-22, CRP, IGF-II, TNF-α, TGF-β, EGFR, MMP-2 and CD-44 Improvement in SGPT and γ-GT levels	[61]
	62	303 mg bid. p.o.; 6 months	Vitamin D 10 mg and Vitamin E 15 mg	Decrease in levels of TBARS, SGPT, HOMA-IR, TNF-α and CRP Elevation in plasmatic levels of estrogens	[298]
NASH	64	210 mg/day p.o.; 8 weeks	—	Reduction in BMI and the level of SGPT and SGOT	[299]
	100	700 mg tid; 48 weeks	—	Decrease in fibrosis Reduction in levels of SGPT and SGOT	[300]
	116	420 and 700 mg tid, p.o.; 48 weeks	—	Improves fibrosis	[301]
Multiple sclerosis therapy-induce liver damage	54 (diagnosed with remitting relapsing multiple sclerosis)	420 mg, p.o.; 6 months	IFNβ	Reduction in SGPT, SGOT, L-17 and IFNγ Decrease in Th1 and Th17 cell population and increase in Treg cell population Increase in IL-10 and TGF-β levels	[302]
Chronic HCV infection	64	47 mg p.o; 12 months	Ribavirin+Peg–IFN and Vitamin E+ phospholipids	Significant decrease in viral load and reduction in plasma markers of liver fibrosis	[236]
	26	5, 10, 15, and 20 mg/kg/day i.v.; 7 and 14 days	Ribavirin+ Peg–IFN	Reduction in HCV RNA production	[229]
	154	420 and 700 mg tid p.o.; 24 weeks	—	Reduction in SGPT levels No change in HCV RNA levels	[56]
HIV/HCV coinfection	16	20 mg/kg/day i.v.; 14 days	Ribavirin+ Peg–IFN and Telaprevir for 12 weeks	Reduction in HCV RNA production	[260]
Anti TB drug-induced hepatotoxicity	55	140 mg tid p.o.; 8 weeks	Rifampicin 10 mg/kg/day, Isoniazid 5 mg/kg/day, Ethambutol 15 mg/kg/day or Pyrazinamide 25 mg/kg/day	Decrease in SGPT, SGOT, γ-GT, ALP and total protein levels	[303]
	70	140 mg tid p.o.; 2 weeks	Isoniazid 5 mg/kg, Pyrazinamide 20 mg/kg, Ethambutol 15 mg/kg and/or Rifampin 10 mg/kg	No significant hepatoprotective effect	[304]
	108	140 mg bid p.o.; 8 weeks	Isoniazid, Pyrazinamide, Ethambutol and/or Rifampin	No significant hepatoprotective effect	[305]
Beta thalassemia	49	140 mg tid p.o.; 9 months	Desferrioxamine	Decrease in serum iron levels and total iron-binding capacity	[306]
	25	420 mg/ day p.o.; 12 weeks	Desferrioxamine 40 mg/kg/day	Reduction in TNF-α and serum neopterin levels Increase in IFNγ and IL-4 production	[307]
	40	140 mg tid p.o.; 6 months	Deferasirox	Decrease in serum ferritin levels	[308]
	22	420 mg/ day p.o.; 6 months	Desferrioxamine	Decrease in TGF-β, IL-23, IL-17 and IL-10 levels	[309]
	80	420 mg/ day p.o.; 9 months	Deferiprone	Decrease in serum ferritin and iron level No change in blood urea, bilirubin, SGPT, SGOT or creatinine levels	[310]

Notably, silymarin has also been reported as a potential candidate for the treatment of alopecia. An in vitro study on human dermal papilla cells conducted by Cheon et al. (2019) demonstrated that therapy with silybin augments the spheroid formation of dermal papilla cells and induces hair-growth properties by triggering AKT and Wnt/β-catenin signaling pathways [283]. Silymarin also plays a crucial role in wound repair and healing. Oryan et al. (2012) reported that silymarin lessens lymphocyte and macrophage cell counts along with elevation in the number of fibrocytes, collagen fibers and fibroblasts [286]. Indeed, regulation of inflammation and oxidation are also essential during wound healing. Studies reported that treatment with silymarin and its active constituents considerably reduces IL-6 and IL-8 discharge and up-regulates IL-8 mRNA expression via NF-κB and AP-1 activation [279]. Furthermore, COX and NOS also help in the progression of wound healing, and silymarin can elevate NOS expression while reducing COX-2 mRNA production [284,285].

## 3. Conclusions

Findings from this review indicate that silymarin is a multifunctional extract which possesses the competency to modulate various cell signaling pathways and induce diverse therapeutic activities. Silymarin is a whole mixture containing different compounds, including silybins A and B, isosilybins A and B, silychristin and silydianin. Silybin is quantitatively the main component of silymarin. Thus, the literature has mainly focused on this compound while ignoring all other components. This leads to problems in reproducibility of the scientific results. Thus, further studies should individually address the main constituents of this mixture which are responsible for the biological activity and determine potential neutral, synergistic and antagonistic effects between these compounds. The inclusion of purified constituents from the silymarin mixture is needed to clarify the bioactivities of the respective compounds in future studies. Moreover, recent advancements in isolation and identification of phytochemicals have also drawn increased attention to the application of herbal medicines as potential targets for the management of various diseases. Silymarin portrays broad anti-inflammatory, antioxidant and pro-apoptotic properties (Figure 7) and modulates various transcription factors (NFκB, PPAR-γ, Nrf2, β-catenin, AP-1, WT-1, kLF6, IRS-1, SREBP-1c, CREB and GADD45α), growth factors such as BDNF, TGF-β and VEGF, receptors (LDL, estrogen receptor, GLP-1, Farnesyl x and Chemokine 4 and 5), signaling pathways (MAPK/ERK2/p53, Slit-2/Robo, Notch, CDK, Wnt/ β-catenin P13K-PKB/Akt, mTOR, IRS-1/P13K/Akt and Jak-STAT), gene expression of apoptotic proteins (Bax, Bcl-2, Bcl-XL, Bim, Caspase 3, 8, 9, FADD and Survivin) and inflammatory cytokines (IL-1β, 2, 5, 6, 8, 12, TNFα, IFNγ, MIPα and MCP-1) while impairing several enzymes (COX-2, iNOS, SGPT, SGOT, MMP, MPO, AChE, G6Pase, MAO-B, LDH, Telomerase, FAS and CK-MB) and activating endogenous antioxidant enzymes, which are consequently accountable for the numerous biological and pharmacological activities reported for silymarin, including hepatoprotection, neuroprotection, cardioprotection and anti-cancer, anti-viral and anti-diabetic properties as evidenced through numerous studies and experimental data.

Therefore, silymarin may be employed as a potential candidate for managing and treating various diseases as a complementary and alternative medicine.

## Figures and Tables

**Figure 1 molecules-27-05327-f001:**
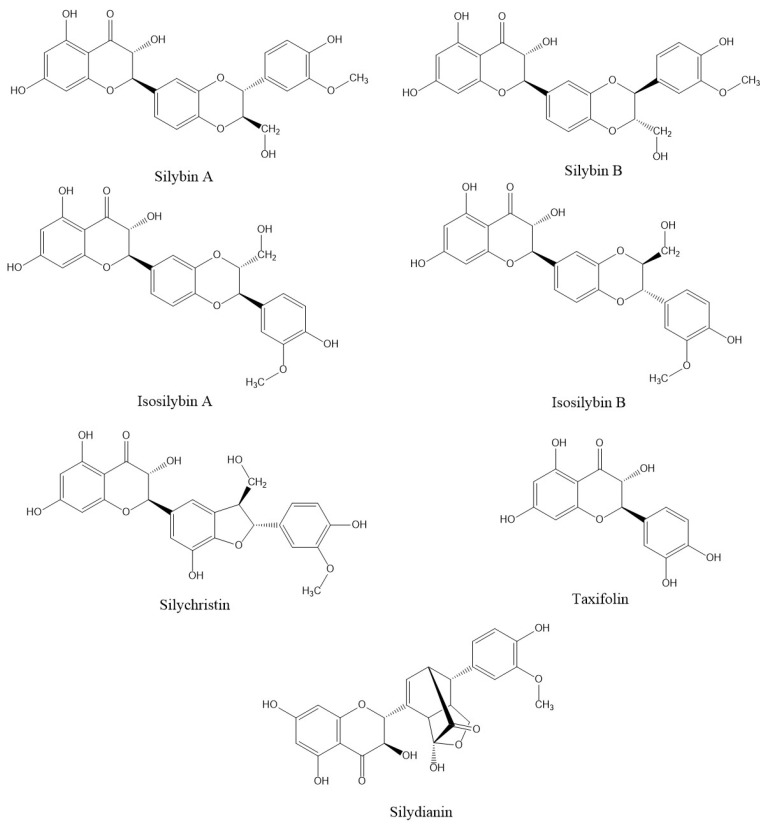
Chemical structures of the phytoconstituents present in silymarin.

**Figure 2 molecules-27-05327-f002:**
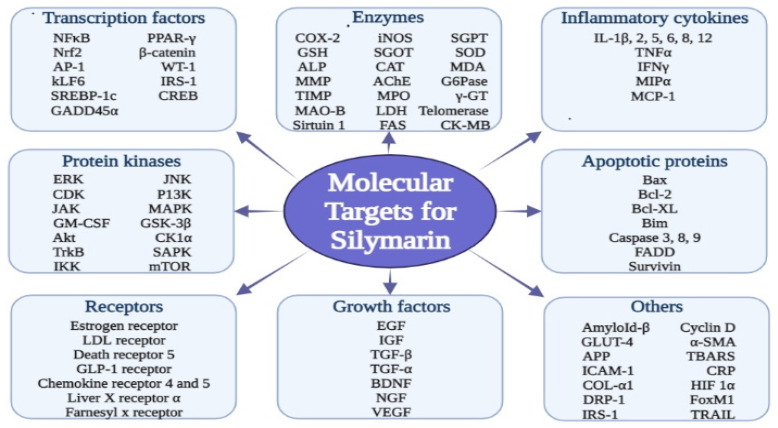
Various molecular targets for silymarin.

**Figure 3 molecules-27-05327-f003:**
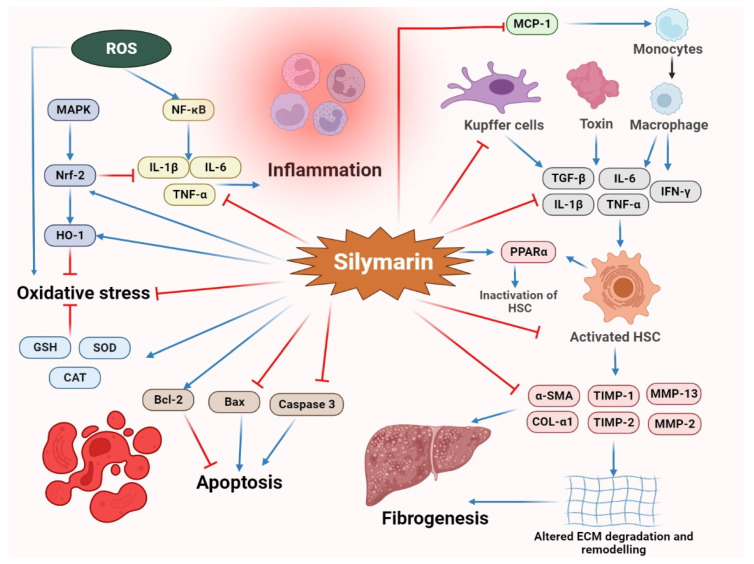
Various hepatoprotective modes of action of silymarin.

**Figure 4 molecules-27-05327-f004:**
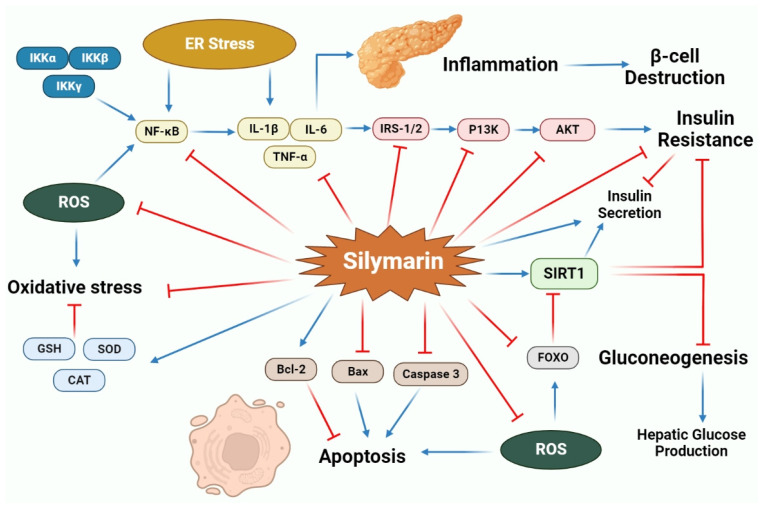
Various anti-diabetic mechanisms of silymarin.

**Figure 5 molecules-27-05327-f005:**
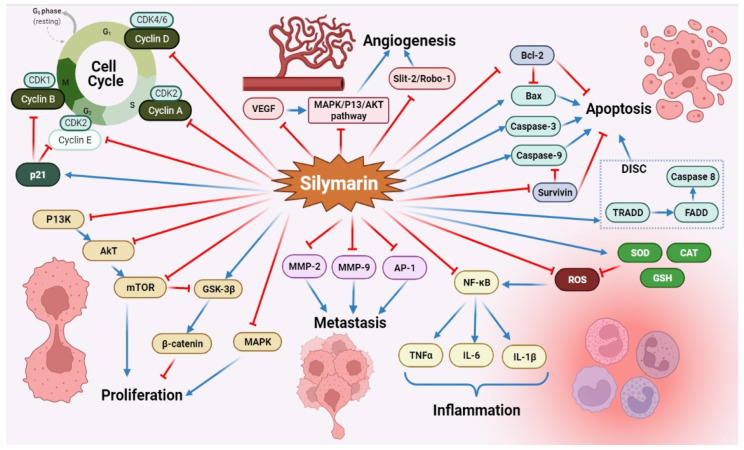
Anti-cancer mechanisms of silymarin.

**Figure 6 molecules-27-05327-f006:**
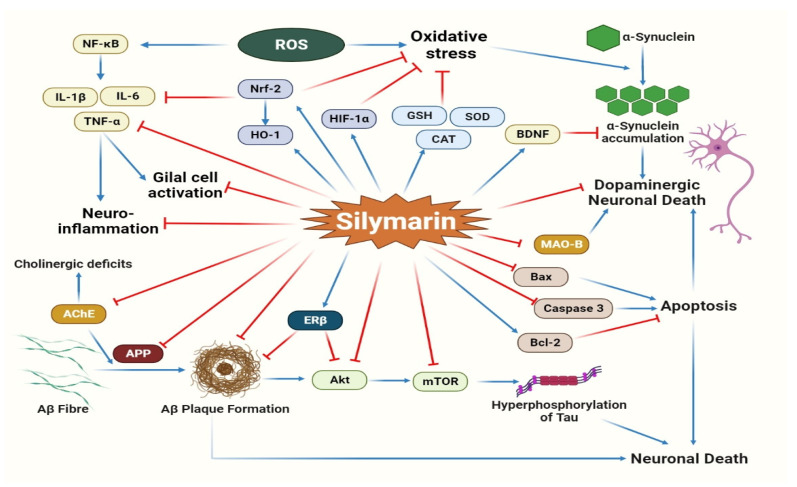
Various mechanisms responsible for the neuroprotective effects of silymarin.

**Figure 7 molecules-27-05327-f007:**
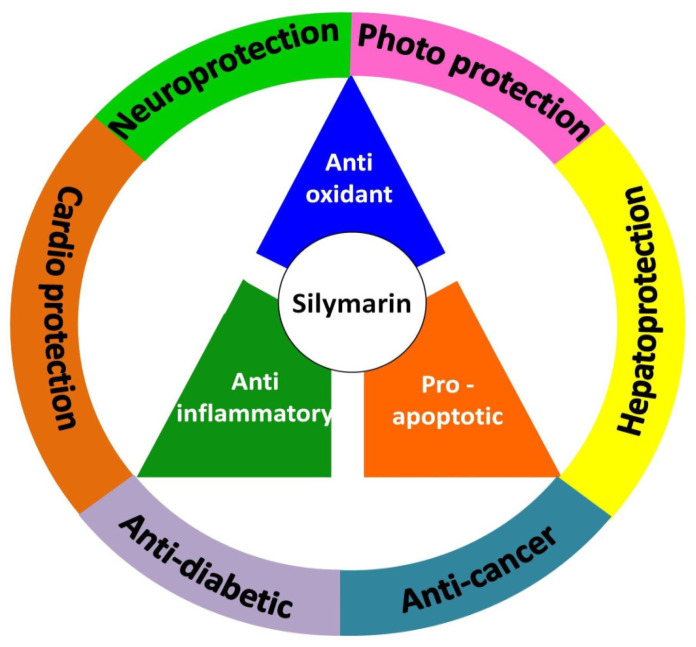
Functional triad of silymarin and its associated pharmacological properties.

**Table 1 molecules-27-05327-t001:** Experimental hepatoprotective activity of silymarin.

Study Model	Dose/Concentration Used	Possible Target Site/Mechanism of Action	Reference
CCl_4_-induced hepatotoxicity	200 mg/kg p.o	Decrease in the levels of ALP, SGPT, and SGOT Reverses the altered expressions of α-SMA	[41]
CCl_4_-induced hepatotoxicity	200 mg/kg p.o	Reduction in the levels of γ-GT, SGPT, SGOT, ALP, TGF-β1, IL-6 and hydroxyproline Down-regulation of α-SMA expressions	[42]
Valproic acid-induced hepatotoxicity	25 and 50 mg/kg	Reduction in the levels of LDH, SGPT, SGOT, ALP Increase in GSH levels	[43]
Thioacetamide-induced hepatotoxicity	100 mg/kg p.o	Down-regulation of TGF-β, AP-1, α-SMA, MMP-2 and 13, COL-α1, TIMP-1 and 2 and *KLF6* expressions	[44]
Fructose-induced NAFLD	400 mg/kg/day p.o.	Reduction in MDA, SGPT, SGOT, hepatic TG, CH and LDL	[39]
Diclofenac-induced hepatotoxicity	200 mg/kg p.o.	Decrease in the levels of MDA, ALP, SGPT, SGOT and TNF-α Elevation in the level of SOD, GSH and CAT	[45]
CCl_4_-induced hepatotoxicity and HSC cells	20 and 100 mg/kg p.o.	Down-regulation of MCP-1, TGF-β and collagen 1 expression	[46]
CCl_4_-induced hepatotoxicity	100 mg/kg i.p.	Reduction in the levels of MDA, GSH, LDL, TG, SGPT, SGOT and ALP	[47]
CCl_4_-induced hepatotoxicity	100 mg/kg i.p. 5 times a week for 4 weeks	Decrease in TG, CH, VLDL-C, ALP, SGPT and SGOT levels Increase in the levels of SOD, GSH and GST Reduction in levels of TBARS, TGF-β1, TNF-α, IL-6, hydroxyproline and resistin	[40]
Acetaminophen-induced hepatotoxicity	200 mg/kg p.o.	Decrease in the levels of SGPT and SGOT Elevation in γ-GT and MPO levels	[48]
NASH rats	200 mg/kg p.o	Reduction in the levels of serum insulin, HOMA-IR, SGOT, SGPT, LDL, TG and TNF-α	[49]
HFD-induced NAFLD	5–10 mL/kg p.o. for 8 weeks	Elevation in the levels of SOD, CAT and PPARα Reduction in levels of MDA, TNF-α, IL-6, SREBP-1c, FAS and LXRα	[50]
MCD diet-induced NASH	105 mg/kg/day p.o. for 8 weeks	Up-regulation of the Nrf2 pathway Decrease in the levels of TNF-α, IL-6, IL-1β, IL-12β, p-IKKα/β, p-*IkBα* and p-*p65* Down-regulation of the NF-κB pathway	[51]
MCD diet-induced NASH	—	Reduction in the levels of SGPT and SGOT Increase in TNF-α, TGF-β and MDA levels Modulates *caspase-3* activation	[52]
Restraint of stress-induced acute liver injury	100 mg/kg	Decrease in MDA and 4-HNE levels Inhibition of JNK activation Decrease in the mRNA levels of IL-1β, IL-6, TNF-α and CCL2 Down-regulation of *Bid, Ba**x* and *caspase-3* and *8*, as well as PARP cleavage	[53]
HepG2 cells and HFD-induced liver inflammation	50 or 100 mg/kg per day	Inhibition in colocalization of NLRP and α-tubulin Down-regulation of cleaved *caspase-1* and thioredoxin-interacting protein Prevents release of IL-1β	[54]
—	600 mg/kg per day p.o. for 10 days	Reduction in the levels of c-Kit, c-Myc, Oct3/4 and SSEA-1 markers Decrease in the levels of MDA, SGPT, SGOT and MPO	[55]
HepG2 cells (Benzo[a]pyrene-induced hepatotoxicity)	0–40 µM	Up-regulation of Nrf2 and PXR Prevents DNA damage	[56]
CCl_4_-induced hepatoxicity	50 and 200 mg/kg	Reduction in TGF-β and α-SMA expression Decrease in the levels of hyaluronic acid Suppresses Kupffer cells activation	[57]
Zidovudine and isoniazid-induced liver toxicity	100 mg/kg	Elevation in the levels of SOD, CAT Reduction in the levels of MDA, SGPT, SGOT and ALP	[58]

**Table 3 molecules-27-05327-t003:** Experimental anti-cancer activity of silymarin.

Type of Cancer	Study Model	Dose/Concentration Used	Possible Target Site/Mechanism of Action	Reference
Bladder cancer	T24 and UM-UC-3 cells	10 μm	Down-regulation of the actin cytoskeleton and PI3K/Akt pathway	[120]
Breast cancer	MDA-MB-231 and MCF-7 breast cancer cells in vivo xenograft tumor model	0–200 25 μg/mL and 50 mg/kg	Reduction in the levels of Bcl-2, p-38 and p-ERK1/2 Elevation in Bax, cleaved poly-ADP ribose polymerase, cleaved caspase-9, and JNK level	[121]
	MCF-7 cells	10–100 μM	Inhibition of BCRP mRNA expression and cell viability	[122]
	4T1 tumor-bearing BAlB/c mice and myeloid-derived suppressor cells	150 mg/kg	Reduction in TNF-α, IL1β and CCR2 levels Improved T cell count	[123]
	MCH-7 and MDA-MB-231 cells	30–90 μM 150–250 μM	Reduction in MMP-2 and 9 protein expression Elevation in E-cadherin expression and reduction in N-cadherin expression Inhibition of NLRP3 inflammasome activation	[124]
	MDA-MB-231 cells	0–400 μM	Reduction in the expression of Cdc42 and D4-GDI mRNA	[125]
	MDA-MB-231 cells	50–350 μM	Inhibition of MMP-2 via inhibition of STAT3	[126]
	MCH-7 cells	—	Reduction in AP-1 dependent MMP-9 gene expression	[127]
	MCF-7 cells	—	Down-regulation of MMP-9 and VEGF expression	[128]
	MDA-MB-231 and T-47D	—	Reduction in cytosolic free β-catenin level Down-regulation of *LPR6* and *Axin2* expression	[129]
Colorectal cancer	Azoxymethane-induced colon carcinogenesis model	300 mg/kg p.o. for 7 days	Reduction in the number of preneoplastic lesions Over-expression of *Bax* protein level Down-regulation of Bcl-2 protein level and IL1β, TNF-α and MMP-7 gene expression	[130]
	SW480 and SW620 cells	300 μM	Elevation in death receptor 4/5 mRNA expression Activation of *caspase-9* Increase in expression of *TRAIL*	[131]
	HCT116 and SW480 cells	0–200 μg/mL	Downregulation of CD1 levels	[132]
	HCT116, SW480, LoVo and HT-29 cells	—	Inhibition of p38, ERK1/2 and GSK3β protein expression	[133]
	Xenograft tumor model	—	Inhibition of the PP2Ac/ AKT Ser473/mTOR pathway Inhibition of cancer stem-like cell development	[134]
Gastric cancer	ASG human gastric cancer cells In vivo xenograft tumor model	20–120 μg/mL and 100 mg/kg	Reduction in the levels of Bcl-2 and p-ERK1/2 Elevation in Bax, cleaved poly-ADP ribose polymerase, cleaved caspase-9, p-P38 and JNK level	[135]
	BGC-823 cells	0–200 µM	Elevation in the levels of *Bax* Activation of *caspase-3*	[136]
	MGC803 cells	0–200 µM	Increase in *caspase-3* and *9* expression Inhibition of p-STAT3, CDK1 and Cyclin B1 protein expression Reduction in *Mcl-1, Bcl-xL* and *survivin* levels	[137]
Hepatocellular carcinoma	Hep G2 cells	0–200 μM	Increase in ceramide secretion Elevation in miRNA levels	[138]
	HCC cells	—	Inhibition of EGFR-dependent Akt signaling	[139]
	Hep G2 cells	12.1–482.4 µg/mL	Decrease in the levels of CXC receptor-4 protein Down-regulation of the Slit-2/Robo-1 pathway	[140]
	Hep G2 cell model and tumor xenograft model	50–200 µM	Elevation in the apoptotic index and *caspase-3* activity Down-regulation of Bcl-2, *survivin* and cyclin D1 level Reduction in Notch1 intracellular domain (NICD), RBP-Jk and Hes1 protein expression	[141]
	*N*-nitrosodiethyl-amine-induced liver cancer	1000 ppm	Inhibition of the recruitment of mast cells Reduction in MMP-2 and 9 expression	[142]
Laryngeal carcinoma	Hep 2 cells	60–300 μM	Down-regulation of *survivin* expression	[143]
Leukemia	K562 cells	0–100 μg/mL	Inhibition of telomerase activity	[144]
Lung cancer	LA795, NCI-H1299 cells and tumor xenograft models	100 mg/kg	Inhibition of *TMEM16A* Reduction in vimentin, N-cadherin and β-catenin levels Elevation in E-cadherin levels Down-regulation of CD1 expression	[145]
Oral cancer	HSC-4, YD15 and Ca9.22 cells tumor xenograft models	40–80 μg/mL 200 mg/kg/day p.o. for 5 weeks	Elevation in the expression of death receptor 5 and cleaved caspase-8 levels	[146]
	MC3 and HN22 cells	—	Elevation in Bim expression Reduction in ERK1/2 levels	[147]
	SCC-25 cells	50 and 100 μM	Reduction in *Bcl-2* gene expression Over-expression of *Bax*, *caspase-3* and *caspase-9* genes	[148]
Ovarian cancer	A2780s and PA-1 cells	50 and 100 µg/mL	Amplification of p53, p21, p27 and Bax protein expression Decrease in Bcl-2 and CDK2 protein expression Activation of caspase-9 and 3	[149]
Prostate cancer	PC-3 and DU-145 cells	—	Reduction in cytosolic free β-catenin levels Down-regulation of *LPR6* and *Axin2* expression	[129]
	DU-145 cells	15.6 to 1000 μM	Activation of SLIT2 protein Down-regulation of CXC receptor 4 expression	[150]
Skin cancer	DMBA–TPA-induced skin papilloma and A431 cells	12.5–50 µM	Reduction in MAPK/ERK1/2 levels and up-regulation of JNK1/2 and *p38* expression	[151]
	A375 and Hs294t cells	0–40 μg/mL	Reduction in β-catenin, MMP-2 and MMP-9 levels Elevation in CK1α and GSK-3β levels	[152]
	DMBA-TPA 2-stage skin carcinogenesis	9 mg topically	Down regulation of NO, TNF-α, IL-6, IL -1β, COX-2, iNOS and NF-κB	[153]
	A375 and Hs294t cells tumor xenograft models	0–60 μg/mL and 500 mg/kg	Up-regulation of *Bax* protein expression Reduction in VEGF, CD31, Bcl-2 and Bcl-xl protein expression Reduction in MMP-2, PCNA and CDK levels	[154]
—	MCF-7 and NCIH-23 cell lines	12.5–200 µg/mL	Up-regulation of *caspase-3, p53* and APAF gene expression	[155]
	*N*-Butyl-*N*-(4-hydroxybutyl) nitrosamine-induced carcinogenesis	1000 ppm	Down-regulation of cyclin D1 expression causing G1 cell arrest	[156]
	U266 MM cell	50–200 μM	Reduction in p-Akt, PI3K and p-mTOR protein expression	[157]

**Table 4 molecules-27-05327-t004:** Cellular pathways modulated by silymarin and its flavonolignans to induce anti-cancer activity.

Type of Cancer	Cellular Pathway Modulated	References
Bladder cancer	↓ PI3K-PKB/Akt signaling pathway	[120]
Cervical/ovarian cancer	↓ MAPK/ERK1/2 and MAPK/p38 signaling pathway ↓ Bcl-2-mediated anti-apoptosis	[158]
Prostate cancer	↓ CDK, MAPK/ERK1/2, and Wnt/β-catenin signaling pathway	[129,159,160]
Skin cancer	↑ p53-mediated apoptosis and MAPK/*p38* signaling pathway ↓ MAPK/ERK1/2, MAPK/JNK1 and Wnt/β-catenin signaling pathway	[151,161,162,163]
Lung cancer	↑ Multiple MAPK signaling pathways ↓ CDK signaling pathway	[145,164]
Liver cancer (Hepatocellular carcinoma)	↓ *Bcl2*-mediated anti-apoptosis ↑ *p53*, Bax and APAF-1-mediated apoptosis ↓ Slit-2/Robo-1 pathway and Notch pathway	[140,141,165]
Breast cancer	↓ MEK/ERK and Wnt/β-catenin signaling pathway ↓ Bcl-2-mediated anti-apoptosis	[121,128,129]
Oral cancer	↑ Bim-mediated apoptosis ↓ MAPK/ERK1/2 signaling pathway	[147]
Colorectal cancer	↓ PP2A/AKT/mTOR, MAPK/ERK1/2 and MAPK/p38 signaling pathway ↓ Bcl-2-mediated anti-apoptosis ↑ Bax-mediated apoptosis	[130,133,134]
Gastric cancer	↓ MAPK/ERK signaling pathway ↑ MAPK/p38 signaling pathway	[128,135]
Peripheral blood cancer	↓ PI3K-PKB/Akt signaling pathway ↑ Caspase-3-mediated apoptosis	[166]

## Data Availability

All the associated data are available within the manuscript.

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
