# Peer review of "Mechanistic Insights into the Pharmacological Significance of Silymarin"

_molecules, 2022, doi:10.3390/molecules27165327_

Round 1

Reviewer 1 Report

This article is well structured and easy to read. For a better understanding of the antidiabetic effects, it would be appropriate to make a figure of the antidiabetic activity as figure 3 related to the anticancer activity. Additionally, there are some points that should be clarified/commented on. There are some adverse reactions when certain drugs are administered with silamalrin but they are not mentioned in the review. One point that I think needs to be commented on is that despite its therapeutic activities, silamyrin is not considered a drug by the FDA, but simply a dietary supplement. Another point concerns the commercial offers of silymarin nanoparticles. Nothing is mentioned about this pharmaceutical formulation which is commercially available.

Author Response

Response to Reviewer's Comments 

Authors responses

The authors are highly thankful to the reviewers for assessing the submission. The suggestions are valuable and have helped us in rectifying the errors in the submission and consequently improving the quality of manuscript. Authors do appreciate all the comments raised by reviewers and editors. All the queries have been addressed, point-by-point in the revised manuscript.

Keeping this in view, we shall be grateful if this revised manuscript could receive your consideration for publication.  The required changes are made in the “Track Change Format”.

#Reviewer 1.

Comment 1.  This article is well structured and easy to read. For a better understanding of the antidiabetic effects, it would be appropriate to make a figure of the antidiabetic activity as figure 3 related to the anticancer activity.          

Response 1. Thank you for this positive appreciation of our manuscript. A New Figure (Figure 4) has been incorporated in the revised manuscript.

Figure 4: Various anti-diabetic mechanisms of silymarin

Comment 2.  Additionally, there are some points that should be clarified/commented on. There are some adverse reactions when certain drugs are administered with silamalrin but they are not mentioned in the review.

Response 2. Thank you for your valuable suggestion. The required information was updated in the manuscript with some references as follows”:

“It has been reported that co-administration of silymarin with some therapeutic agents may enhance its biological activity that is reduced by the liver, such as amitriptyline, diazepam, celecoxib, fluvastatin, diclofenac, zileuton, ibuprofen, glipizide, losartan, irbesartan, piroxicam, tolbutamide, torsemide, tamoxifen, phenytoin Han et al., reported that coadministration of different sources of silymarin with losartan significantly improved the systemic concentration of losartan. Molto et al., examined the effect of the co-administration of silymarin with darunavir and ritonavir combination in HIV patients. They reported a decline in AUC and Cmax, when co-administration of silymarin with combination of drugs compared to combination of drugs alone. “

Comment 3.  One point that I think needs to be commented on is that despite its therapeutic activities, silamyrin is not considered a drug by the FDA, but simply a dietary supplement.

Response 3. Thank you for your valuable suggestion. The authors have incorporated this statement in the Introduction section.

Comment 4.  Another point concerns the commercial offers of silymarin nanoparticles. Nothing is mentioned about this pharmaceutical formulation which is commercially available.

Response 4. Thank you for your valuable suggestion. A short paragraph has been incorporated to address this query with addition of some marketed formulations available in the market. The paragraph is:

“Silymarin offers several benefits in contrast to other therapeutic agents because of its non-toxicity and high hydrophobic properties. Although, it has low aqueous solubility that results in poor bioavailability. This issue can be resolved by employing nanosystem-based approaches. Nanoparticles are natural or chemically synthesized particles having particle size range 1- 200 nm. Because of their small size range, they offer various advantages such as enhanced interaction area, enhanced aqueous solubility and intracellular permeability. Apart from this, they can reduce multi-drug resistance of many anti-cancer agents including silybin. These are distributed across the body depending upon various factors such as their small size, which aids in longer systematic circulation periods and their potential to take advantage of anti-cancer properties. They also show greater stability during storage. Nanoparticles and their use in drug delivery are a much more efficient approach for cancer treatment than traditional chemotherapy. There are as many as 75 brands of silymarin available in market in different dosage forms such as tablets, capsules, syrups, etc. with an enhanced bioavailability under the trade names like Livergol, Silipide, Carsil tablets, Legalon capsules, Alrin-B syrup etc. Nano silymarin OIC, approved by Vietnam Drug Administration, is the only patented nanoformulation that is commercially available as a dietary supplement, in form of capsules, for improving liver function.”

Reviewer 2 Report

This is a thorough and necessary investigation of one of the most important medicinal plants.

The manuscript is well-written and informative. The extensive literature review makes it an extremely useful work for pharmacognosy researchers.

My only (minor) concern is the authors' lack of critical perspective on the difficulties in identifying the compound(s) responsible for the observed activities. Indeed, most studies used silymarin mixtures that were not always characterized from a phytochemical standpoint, making it difficult to propose a clear mode of action, as the group of V. Kren did for antioxidant activities that were well described from a structure-activity standpoint. This point, in my opinion, should be critically discussed (the inclusion of purified compound from the silymarin mixture in future studies to clarify the compound responsible for the activity as well as determined potential neutral, synergy, antagonistic effect between these compounds).

There are also some minor corrections that need to be completed before publication:

Fruits and akenes, not seeds (abstract)

silybins has to be used instead of "silybin" to referred to the mixture composed of silybin A, silybin B, isosilybin A and isosilybin B (eg in the abstract)

To the best of my knowledged quercetin is not considered a constiuent of the silymarin mixture (eg in the abstract).

"has been widely employed for several decades as a natural medication for various liver diseases" In fact silymarin, preparation of silybum marianum fruit, usage is described in the first european pharmacopeae for liver diseases, so much more than "several decades". (Introduction)

silibinin A and silibinin B should be silybin A and silybin B (Introduction)

Same remark for isosilybins A and B.

other minor flavonolignans like isosilychristin have to be mentionned.

2. Pharmacological activities. The first mentionned usage (and current pharmaceutical usage) of silymarin is to cure liver diseases. It would be logical to start by this one.

The text can be sometimes shortenned, like the introductive paragraph for Cancer which is in my opinion too general and too long.

Author Response

#Reviewer 2. All the required changes are made in “Track Change” format.

Comment 1. This is a thorough and necessary investigation of one of the most important medicinal plants. The manuscript is well-written and informative. The extensive literature review makes it an extremely useful work for pharmacognosy researchers.

Response 1. Thank you for your positive appreciation of this manuscript.

Comment 2. My only (minor) concern is the authors' lack of critical perspective on the difficulties in identifying the compound(s) responsible for the observed activities. Indeed, most studies used silymarin mixtures that were not always characterized from a phytochemical standpoint, making it difficult to propose a clear mode of action, as the group of V. Kren did for antioxidant activities that were well described from a structure-activity standpoint. This point, in my opinion, should be critically discussed (the inclusion of purified compound from the silymarin mixture in future studies to clarify the compound responsible for the activity as well as determined potential neutral, synergy, antagonistic effect between these compounds).

Response 2. Thank you for your suggestion. This query has been addressed in the conclusion section.

Comment 3. There are also some minor corrections that need to be completed before publication:

Fruits and akenes, not seeds (abstract).        

Response 3. As suggested, the words “seeds and fruits” have been replaced with “fruits and akenes”.

Comment 4. Silybins has to be used instead of "silybin" to referred to the mixture composed of silybin A, silybin B, isosilybin A and isosilybin B (eg in the abstract)    

Response 4. Thank you for your valuable suggestion. The word “silybin” is replaced with “silybins” wherever applicable.

Comment 5. To the best of my knowledged quercetin is not considered a constituent of the silymarin mixture (eg in the abstract).    

Response 5. Thank you for your suggestion. As suggested, quercetin was removed from the list of constituents.

Comment 6. "has been widely employed for several decades as a natural medication for various liver diseases" In fact silymarin, preparation of silybum marianum fruit, usage is described in the first european pharmacopeae for liver diseases, so much more than "several decades". (Introduction).

Response 6. Thank you for your valuable suggestion. As per suggestion, this sentence was modified in the revised version.

Comment 7. Silibinin A and silibinin B should be silybin A and silybin B (Introduction)

Same remark for isosilybins A and B.

Response 7. Thank you for your suggestion. The word “silibinin” was replaced with “silybin” in the entire manuscript. The same was also done for isosilybins.

Comment 8. Other minor flavonolignans like isosilychristin have to be mentioned.     

Response 8. As suggested, flavonolignans were also mentioned in the manuscript.

Comment 9. Pharmacological activities. The first mentioned usage (and current pharmaceutical usage) of silymarin is to cure liver diseases. It would be logical to start by this one.

Response 9. Thank you for your valuable suggestion. The suggested changes were made in the revised manuscript.

Comment 10. The text can be sometimes shortened, like the introductive paragraph for Cancer which is in my opinion too general and too long.    

Response 10. Thank you for your valuable suggestion. The introductive paragraph for Cancer was reduced a lot in the revised manuscript.